# Impacts of emission reduction and meteorological conditions on air quality improvement during the 2014 Youth Olympic Games in Nanjing, China

Qian Huang [1, 2], Tijian Wang [1, *], Pulong Chen [1], Xiaoxian Huang [3], Jialei Zhu [4], and Bingliang Zhuang [1]

[1] School of Atmospheric Sciences, CMA-NJU Joint Laboratory for Climate Prediction Studies, Jiangsu Collaborative Innovation Center for Climate Change, Nanjing University, Nanjing, 210023, China

[2] Scientific Research Academy of Guangxi Environmental Protection, Nanning, 530022, China

[3] College of Plant Science & Technology, Huazhong Agricultural University, Wuhan, 430070, China

[4] Department of Climate and Space Sciences and Engineering, University of Michigan, Ann Arbor, 48109, USA

Correspondence to: Tijian Wang (tjwang@nju.edu.cn)

**Abstract**

As the holding city of the 2nd Youth Olympic Games (YOG), Nanjing is highly industrialized and urbanized facing with several air pollution issues. In order to ensure better air quality during the event, the local government took great efforts to control the pollution emissions. However, air quality can still be affected by synoptic weather. In this paper, the influences of meteorological factors and emission reductions were investigated using observational data and numerical simulations with WRF/CMAQ. During the YOG holding month (Aug., 2014), the observed hourly mean concentrations of $SO_2$, $NO_2$, $PM_{10}$, $PM_{2.5}$, CO and $O_3$ were 11.6 μg/m$^3$, 34.0 μg/m$^3$, 57.8 μg/m$^3$, 39.4 μg/m$^3$, 0.9 mg/m$^3$, and 38.8 μg/m$^3$, respectively, which were below China National Ambient Air Quality Standard (Level 2). However, model simulation

showed that the weather conditions such as weaker winds during the holding time were adverse for better air quality, and could increase $SO_2$, $NO_2$, $PM_{10}$, $PM_{2.5}$ and CO by 17.5%, 16.9%, 18.5%, 18.8%, 7.8% and 0.8%, respectively. Taking account of local emission abatement only, the simulated $SO_2$, $NO_2$, $PM_{10}$, $PM_{2.5}$ and CO was decreased by 24.6%, 12.1%, 15.1%, 8.1% and 7.2%, respectively. Consequently, stringent emission control measures can reduce the concentrations of air pollutants in short term, and emission reduction is the very important factor for the air quality improvement during the YOG, which has set up a good example in air protection for important social events.

**KEY WORDS:** Youth Olympic Games; Emission reduction; Meteorological conditions; WRF/CMAQ; Nanjing

## 1 Introduction

As located in the economically developed Yangtze River Delta (YRD) region of China, Nanjing successfully hosted the second Youth Olympic Games (YOG) during 16 - 28 Aug., 2014. Nanjing is a highly urbanized city and its GDP ranked the 12th of all the cities in China by 2013 (National Bureau of Statistics of China, 2014). Due to fast urbanization and industrialization, heavy motor vehicles and construction dust, Nanjing has long been suffered from air pollution (Dong et al., 2013; Chen et al., 2015).

In order to realize the promise of "Green YOG", the local government had taken a series of measures to reduce emissions of air pollutants. The preparatory work started from 1 Jul., 2014. Besides, the local government performed the stringent environmental quality assurance work plan from 1 Aug. (National Bureau of Statistics of China, 2014). The controlled emissions include 5 major categories: industry, power plants, traffic, VOC product-related sources and others. Some local petrochemical, chemical and steel industries were forced to limit or halt the production. Coal-combustion enterprises were required to use high-quality coals with low sulfur content and ash content. And vehicles with heavy pollution called "yellow label buses" were prohibited in Nanjing during 10-28 Aug.. Oil loading and unloading

operations were strictly controlled. All construction processes in the city were forced
to stop. Surface with bare soil was covered.

62       It is well known that air quality can be affected by both meteorological factors

and pollutant emissions. Many cases verified that both emission abatement efforts and
weather conditions can influence the improvement of local air quality. Emission
control has been taken in many social events, like Beijing Olympic Games in 2008
and Shanghai Expo in 2010. Xing et al. (2011) suggested that emission controls
benefit for pollutants reduction, but meteorological effects can be either ways at
different locations. Cermak and Knutti (2009), Wang et al. (2009b, 2010) and Xing et
al. (2011) reported that typical meteorological conditions accounted more for air
improvement during 2008 Beijing Olympics than emission reductions. Zhou et al.
(2010) concluded that transportation control measures resulted in a reduction of
44.5% and 49.0% in daily CO and $NO_x$ emission from motor vehicles during the 2008
Olympics. Cai et al. (2011) and Wang et al. (2009a) also studied the transportation
controls on improving air quality during Beijing Olympic Games. Okuda et al. (2011)
argued that sources control during Beijing Olympics significantly reduced $PM_{10}$, $NO_2$
and $SO_2$, but did not as effectively reduce $PM_{2.5}$. Streets et al. (2007) proposed that
local sources controlling is inadequate for heavily populated, urbanized, and
industrialized city, regional air quality management is in urgent need. Lin et al. (2013)
applied monitoring data to analyze the weather impacts on air quality of the World
Expo in YRD and concluded that high frequency of marine winds during the Expo
had a positive effect on the air quality of coastal cities, but a negative effect on some
inland cities in YRD. Satellite data reflected that the tropospheric $NO_2$ column,
aerosol optical thickness (AOT), and CO concentration dropped by 8%, 14% and 12%,
respectively over Shanghai during the Expo period, compared to the past three years
(Hao et al., 2011). Liu et al. (2013) compared the contributions of long-term and
short-term emission control via CMAQ simulation and compared their effects on air
quality in Guangzhou during the Asian Games. Xu et al. (2013) concluded that $PM_{2.5}$
was mainly emitted from anthropogenic sources other than biogenic sources and
indicated that cutting down anthropogenic emissions could decrease $PM_{2.5}$ effectively.
Dong et al. (2013) found that independent $NO_x$ emission reduction would strengthen
$O_3$ as a side effect in YRD. Chen et al. (2015, 2017) studied the source apportionment
of size-fractionated particles in Nanjing, and found that construction dust contributes
the most in coarse particles, and fugitive and construction dust decreased significantly
in YOG.

95        There have been some studies on air quality during the 2nd YOG (Ding et al.,

2015; Chen et al., 2017; Zhou et al. 2017), but few work focused on the relative
contributions of meteorology and control efforts. This study takes the air quality
monitoring data and applies WRF/CMAQ model to estimate the effect of
meteorological factors and emission reduction on air quality of Nanjing during the
2nd YOG. Data and model descriptions as well as simulation scenarios are described
in Section 2. Section 3 examines the characteristics of six major air pollutants ($SO_2$,
$NO_2$, $PM_{10}$, $PM_{2.5}$, CO and $O_3$) and compares their concentrations during the YOG
with those a year ago and the months without emission reduction (Jul. and Sept.,
2014). Besides, this section discusses the separate effect of weather conditions and
emission abatement qualitatively and quantitatively based on the simulation results.
Section 4 summaries the main conclusions, emphasizes the important factor of the air
quality promotion during the YOG, and provides some advice for ensuring pleasant
future air quality.

**2 Methodology**
2.1 Data description

112       Hourly observed air quality data during Jul.- Sept. 2014 and Aug. 2013 of two

representative stations was collected from Nanjing Environmental Monitoring Center
(http://222.190.111.117:8023/). Both of the two stations are state controlling air
monitoring sites. The data quality assurance and quality control procedures for
monitoring strictly follow the national standards (State Environmental Protection
Administration of China, 2006). Caochangmen (CCM) Station (118.75° E, 32.06° N)
locates in Gulou District, the city center of Nanjing. Gulou District is the center of
economy, politics, culture and education in Nanjing. Here gathers many East China's
high-end industrial and corporate headquarters. Besides, over 90% provincial
authorities, more than 20 colleges and universities, and more than 120 research
institutes situate in Gulou District. It's the most populated area in Nanjing, with lively
commercial hub and heavy traffic. Thus, CCM station was chosen to represent the
urban status of Nanjing. The other site calls Xianlin (XL) Station (118.92° E, 32.11°
N ), which locates in Qixia District, the suburb of Nanjing. Compared to Gulou
District, Qixia District is much more sparsely populated with few traffic congestion
problem. Thus, XL station was chosen to represent the suburban status of Nanjing.

2.2 Model description

130        The integrated modeling system WRF/CMAQ was employed in this research.

Community Multiscale Air Quality (CMAQ) is a third-generation regional air quality
model developed by the Environmental Protection Agency of USA (USEPA). It
incorporates a set of up-to-date compatible modules and control equations for the
atmosphere, and can fully consider complicated physical and chemical processes
(Byun and Schere, 2006; Foley et al., 2010). Many applications have proven that
CMAQ is a reliable tool in simulating air quality from city scale to mesoscale (Xing
et al., 2011; Dong et al., 2013; Liu et al., 2013; Xu et al. 2013; Shu et al., 2016).
Community Multiscale Air Quality (CMAQ v4.7.1, Binkowski and Roselle, 2003)
model includes the 2005 Carbon Bond gas-phase mechanism (CB05) (Yarwood et al.,
2005) and the fourth-generation CMAQ aerosol module (AERO4) (Byun and Schere,
2006). And it was applied to simulate the pollutant distribution over Nanjing in this
paper. Weather Research and Forecasting (WRF) is a new generation of mesoscale
weather forecast model and assimilation system, developed by the National Center for
Atmospheric Research (NCAR). It has been widely applied in China and shows a
good performance in all kinds of weather forecasts (Jiang et al., 2008, 2012; Xu et
al.,2013; Liao et al., 2014, 2015; Xie et al., 2014, 2016; Li et al., 2016; Shu et al.,
2016). WRF v3.2.1 (Skamarock et al., 2008) model was run to provide meteorology
fields for CMAQ. Four nested domains were set for both models, with horizontal
resolutions of 81km, 27km, 9km, 3km, with the innermost domain covering Nanjing
(Fig. 1). For all domains, 23 vertical sigma layer from the surface to the top pressure
of 100 hpa was set, with about 10 layers in the planetary boundary layer. The detail
dynamic parameterization in WRF as well as the physical and chemical schemes of
CMAQ applied in this research were the same as those in Shu et al. (2016)'s work and
were proven to have good performance. As for the innermost domain, Nanjing
Environmental Protection Bureau chooses the local 9 state controlling air monitoring
sites (See Fig. 1, Table 1) to represent the whole Nanjing (NJ) city. In conformity with
this, the 9 sites in domain4 were chosen to represent the whole Nanjing while
analyzing the impacts of weather conditions and emission reduction.

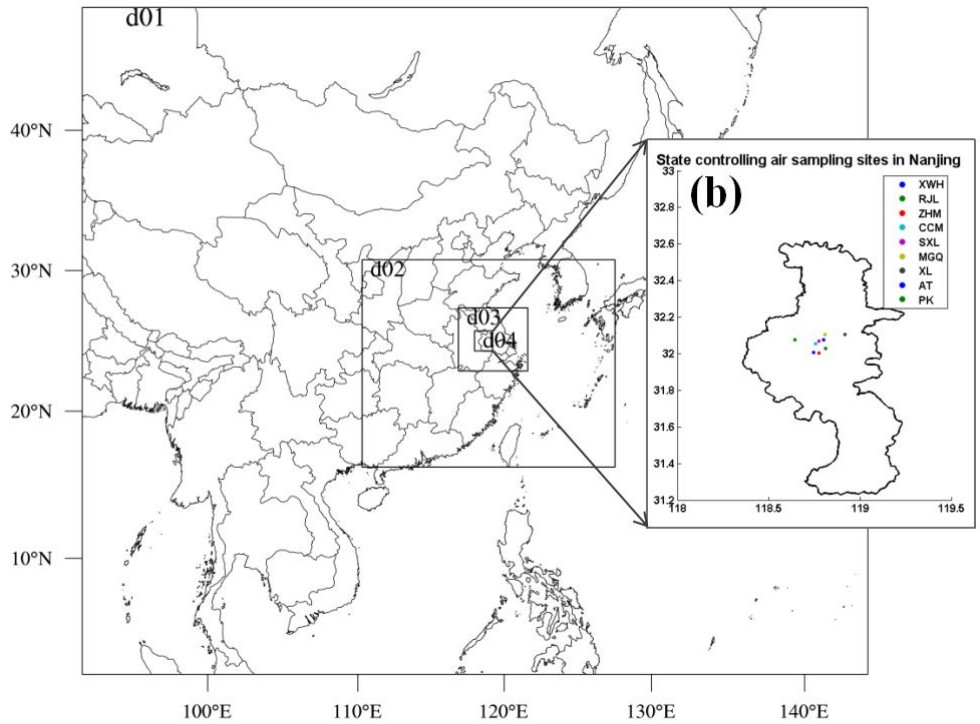


**Fig. 1.** Modeling domains and state controlling air monitoring sites in Nanjing. ((a) The four nested
modeling domains at 81km (d01: East Asia), 27km (d02: East China), 9km (d03: Yangtze River Delta),
and 3km (d04: Nanjing), (b) Locations of 9 sites in Nanjing).

**Table 1**
The air monitoring sites in Nanjing

| Sites | Abbreviations | Location |
|---|---|---|
| Xuanwuhu Station | XWH | 32.08° N, 118.80° E |
| Ruijinlu Station | RJL | 32.03° N, 118.82° E |

| | | |
|---|---|---|
| Zhonghuamen Station | ZHM | 32.00° N, 118.76° E |
| Caochangmen Station | CCM | 32.06° N, 118.75° E |
| Shanxilu Station | SXL | 32.07° N, 118.77° E |
| Maigaoqiao Station | MGQ | 32.11° N, 118.81° E |
| Xianlin Station | XL | 32.11° N, 118.92° E |
| Aoti Station | AT | 32.01° N, 118.74° E |
| Pukou Station | PK | 32.07° N, 118.64° E |


2.3 Emissions and simulation scenarios
In this study, Multi-resolution Emission Inventory for China (MEIC v1.2,
http://www.meicmodel.org/) with a resolution of 0.25° × 0.25° was employed to
provide the anthropogenic emissions for species including $SO_2$, $NO_x$, CO, NMVOC,
$NH_3$, $CO_2$, $PM_{2.5}$, $PM_{10}$, BC, and OC, form 4 sectors: industry, power plants,
transportation, and residential.
For the innermost domain, the local emission inventory before and after emission
control was used with a horizontal resolution of 3km × 3km. The base year of the
local emission is 2012. According to the local emission control program, 5 major
categories: industry, power plants, traffic, VOC product-related sources and others
were in the control list. In Aug. 2014, all coal-combustion enterprises must use
high-quality coals with low sulfur content less than 0.5% and ash content less than
13%. Besides, the local government ordered over 100 local petrochemical, chemical
and steel enterprises to cut or halt their production. Moreover, heavy pollution
vehicles were prohibited in Nanjing during 10-28 Aug. 2014 to reduce traffic
emission. To reduce emissions of volatile organic compounds, loading and unloading
oil operations were prohibited at the docks in Nanjing section of Yangtze River.
What's more, local construction work was halted during Aug. 2014. With these efforts,
the emission would be cut by 25.0% for $SO_2$, 15.0% for $NO_x$, 42.8% for $PM_{10}$, 36.2%
for $PM_{2.5}$, and 20.0% for CO. The spatial distributions of emission reduction were
showed in Fig. 2. For $SO_2$, $NO_x$, $PM_{10}$ and $PM_{2.5}$, the emission reduction area centered
in the middle of Nanjing city. And for CO, the emission reduction centered in several
points.

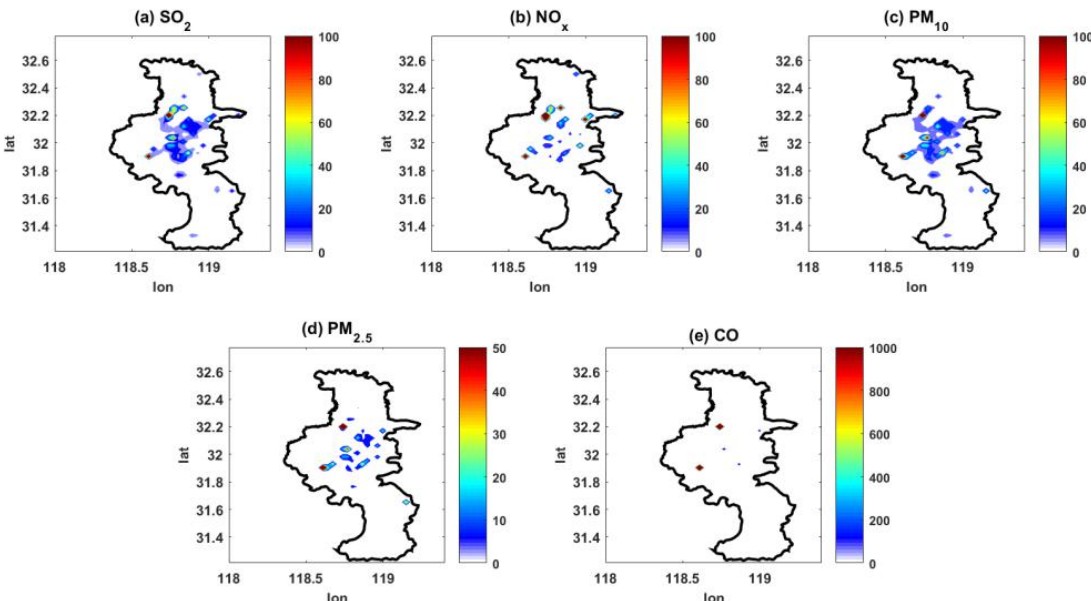

**Fig. 2.** Emission reduction in domain4 ((a) $SO_2$, (b) $NO_X$, (c) $PM_{10}$, (d) $PM_{2.5}$, (e) CO (unit: t/month)).

The simulated period was from Jul. 27 to Sept. 1 (China standard time, CST), but only the holding month (1 Aug. to 31 Aug.) was focused on for discussions. In order to better understand the influence of meteorological factors and emission abatement, three experiments were carried out. Exp. 1 used the weather conditions during Aug. 2014 (CST) and the emission inventory after reduction. Exp. 2 used the same weather conditions as Exp. 1 with the emission inventory before reduction. Exp. 3 used the same inventory as Exp. 2 and the weather conditions during Aug. 2013 (CST). Besides, Exp. 2 acted as the control experiment. Therefore, Exp. 1 and Exp. 2 were performed to investigate the influence of emission reduction on pollutants. Similarly, Exp. 2 and Exp. 3 were conducted to understand the impact of meteorology on air quality.

## 3 Results and discussion

3.1 Observed air quality during the YOG

In the most strictly control month Aug. 2014, emission sources including 5 major categories were reduced, and the air quality had great promotion compared to Aug. 2013. Firstly, it was good during the Games in accordance with China's National Ambient Air Quality Standards (NAAQS) (Ministry of Environmental Protection of

the People's Republic of China, 2012) (Fig. 3, Fig. 4). The hourly mean pollutant concentration of the two sites during Aug. 2014 is 11.6 µg/m³ for $SO_2$, 34.0 µg/m³ for $NO_2$, 57.8 µg/m³ for $PM_{10}$, 39.4 µg/m³ for $PM_{2.5}$, 0.9 mg/m³ for CO, and 38.8 µg/m³ for $O_3$. Secondly, as showed in Table 2 and Table 3, the mean concentration of the six major species dropped by 64.7% for $SO_2$, 29.8% for $PM_{10}$, 9.8% for $PM_{2.5}$, 8.9% for CO and 31.7% for $O_3$ at CCM station, while 50.0% for $SO_2$, 18.6% for $NO_2$, 32.8% for $PM_{10}$, 4.1% for $PM_{2.5}$, and 31.7% for $O_3$ at XL station. Besides, the smaller standard deviation (std) of $SO_2$, $NO_2$, CO and $O_3$ revealed that concentrations of these air pollutants varied more steadily in Aug. 2014. However, the drop of pollutant concentration could be caused mainly by meteorology conditions or emission reductions. And we will discuss this issue based on model simulations in Section 3.2 and Section 3.3.

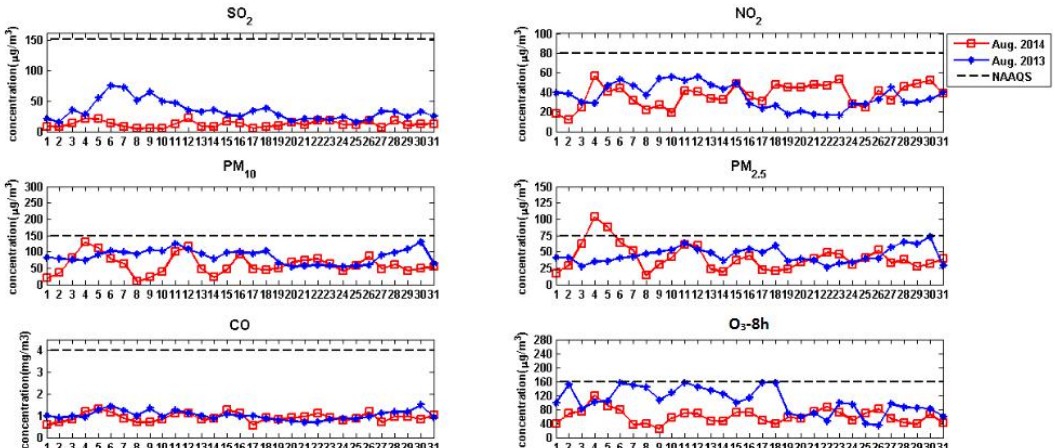

**Fig. 3.** Day-to-day variations in $SO_2$, $NO_2$, $PM_{10}$, $PM_{2.5}$, CO and $O_3$-8h at CCM station in Aug. 2013 and Aug. 2014 (CST). Observed data in Aug. 2013 and Aug. 2014 are indicated in blue and red, respectively. NAAQS are indicated in black dotted line.

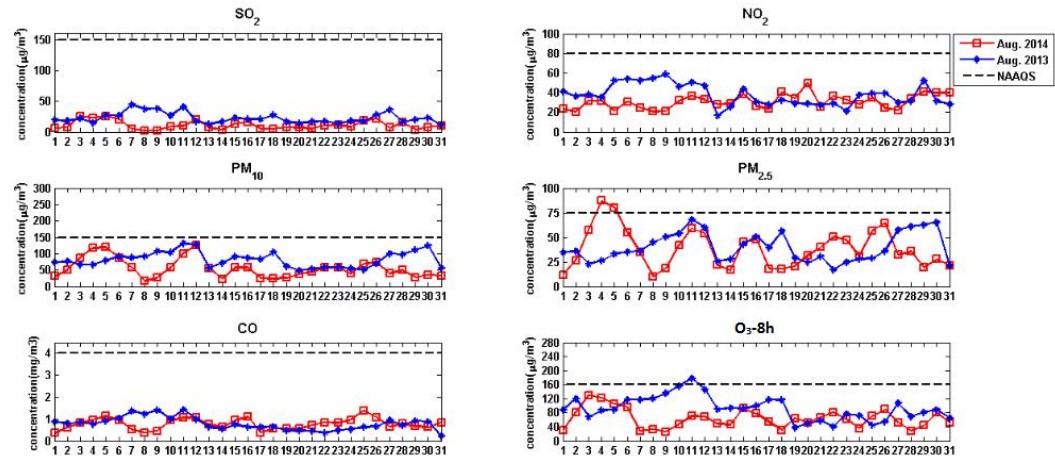

**Fig. 4.** Day-to-day variations in $SO_2$, $NO_2$, $PM_{10}$, $PM_{2.5}$, CO and $O_3$-8h at XL station in Aug. 2013 and
Aug. 2014 (CST). Observed data in Aug. 2013 and Aug. 2014 are indicated in blue and red,
respectively. NAAQS are indicated in black dotted line.

**Table 2**
Statistical analysis of hourly data in Aug. 2013 and Aug. 2014 at CCM station (The unit is µg/m³
except CO (mg/m³))

| species | time | max | min | mean | median | std | Δ |
|---|---|---|---|---|---|---|---|
| $SO_2$ | Aug. 2013 | 169.0 | 1.0 | 33.7 | 27.0 | 23.7 | |
| | Aug. 2014 | 72.0 | 2.0 | 11.9 | 10.0 | 7.8 | -64.7% |
| $NO_2$ | Aug. 2013 | 111.0 | 1.0 | 35.4 | 32.0 | 19.4 | |
| | Aug. 2014 | 110.0 | 1.0 | 37.3 | 35.0 | 18.6 | 5.0% |
| $PM_{10}$ | Aug. 2013 | 213.0 | 19.0 | 86.0 | 84.0 | 29.5 | |
| | Aug. 2014 | 198.0 | 6.0 | 60.4 | 54.0 | 36.6 | -29.8% |
| $PM_{2.5}$ | Aug. 2013 | 123.0 | 10.0 | 45.2 | 43.5 | 16.2 | |
| | Aug. 2014 | 165.0 | 3.0 | 40.7 | 36.0 | 23.8 | -9.8% |
| CO | Aug. 2013 | 3.1 | 0.4 | 1.0 | 0.9 | 0.4 | |
| | Aug. 2014 | 2.2 | 0.3 | 0.9 | 0.9 | 0.3 | -8.9% |
| $O_3$ | Aug. 2013 | 198.0 | 1.0 | 56.9 | 42.0 | 46.2 | |
| | Aug. 2014 | 150.0 | 9.0 | 38.9 | 34.0 | 22.6 | -31.7% |

Δ : the change percentage of species in Aug. 2014 based on Aug. 2013.

**Table 3**
Statistical analysis of hourly data in Aug. 2013 and Aug. 2014 at XL station (The unit is µg/m³ except
CO (mg/m³))

| species | time | max | min | mean | median | std | Δ |
|---|---|---|---|---|---|---|---|
| $SO_2$ | Aug. 2013 | 139.0 | 0.0 | 22.8 | 19.0 | 16.1 | |
| | Aug. 2014 | 71.0 | 1.0 | 11.4 | 8.0 | 10.4 | -50.0% |
| $NO_2$ | Aug. 2013 | 129.0 | 0.0 | 37.7 | 32.0 | 21.7 | |

| | | | | | | | $\Delta$ |
|---|---|---|---|---|---|---|---|
| | Aug. 2014 | 95.0 | 7.0 | 30.7 | 27.0 | 15.0 | -18.6% |
| $PM_{10}$ | Aug. 2013 | 215.0 | 0.0 | 82.1 | 79.0 | 32.4 | |
| | Aug. 2014 | 196.0 | 6.0 | 55.2 | 47.0 | 35.9 | -32.8% |
| $PM_{2.5}$ | Aug. 2013 | 122.0 | 0.0 | 39.7 | 37.5 | 18.9 | |
| | Aug. 2014 | 157.0 | 3.0 | 38.0 | 34.0 | 24.1 | -4.1% |
| CO | Aug. 2013 | 3.2 | 0.0 | 0.8 | 0.7 | 0.4 | |
| | Aug. 2014 | 2.0 | 0.3 | 0.8 | 0.7 | 0.3 | <0.1% |
| $O_3$ | Aug. 2013 | 193.0 | 0.0 | 56.6 | 44.0 | 37.5 | |
| | Aug. 2014 | 148.0 | 2.0 | 38.7 | 32.0 | 28.3 | -31.7% |

$\Delta$ : the change percentage of species in Aug. 2014 based on Aug. 2013.

Analogously, compared the observational data in Aug. 2014 with that in Jul. and Sept. 2014 (the months before and after the most aggressive abatement), the concentrations of most species also decreased obviously. As presented in Fig. 5 and Fig. 6, without abatement, $NO_2$, $PM_{10}$, $PM_{2.5}$ and $O_3$ were likely to exceed NAAQS, especially for $PM_{2.5}$ and $O_3$. As shown in Table 4 and Table 5, compared with Jul. 2014, the concentration of $NO_2$, $PM_{10}$, $PM_{2.5}$, CO and $O_3$ dropped by 0.7%, 31.8%, 33.7%, 1.1%, and 52.8%, respectively at CCM station in Aug. 2014, while the concentration of $SO_2$, $NO_2$, $PM_{10}$, $PM_{2.5}$, CO and $O_3$ decreased by 15.8%, 39.6%, 34.6%, 7.1%, and 50.7%, respectively at XL station in Aug. 2014. Without emission control, the concentration of air pollutants rebounded in Sept. 2014. Compared to Aug., the concentration of $SO_2$, $NO_2$, $PM_{10}$, $PM_{2.5}$ and $O_3$ increased by 37.4%, 19.8%, 37.6%, 22.3%, and 47.2%, respectively at CCM station in Sept. 2014 (Table 4), while the concentration of $SO_2$, $NO_2$, $PM_{10}$, $PM_{2.5}$, CO and $O_3$ increased by 24.6%, 21.8%, 28.7%, 17.7%, 4.9%, and 39.9%, respectively at XL station in Sept. 2014 (Table 5). Besides, for most species, the standard deviation was the lowest in Aug., which meant that the potential of extreme events was the least in Aug.. Assume that the weather conditions in Jul., Aug., Sept., 2014 were similar, it can be estimated that emission sources could be the major impact factor of explaining the concentration changes during the three months. These results proved that concentrations of most species decreased and had less potential in extreme events after aggressive emission abatement. However, the concentration became higher without emission control.

Section 3.3 would further discuss the change of pollutant concentration with and
without emission reduction based on model simulation.

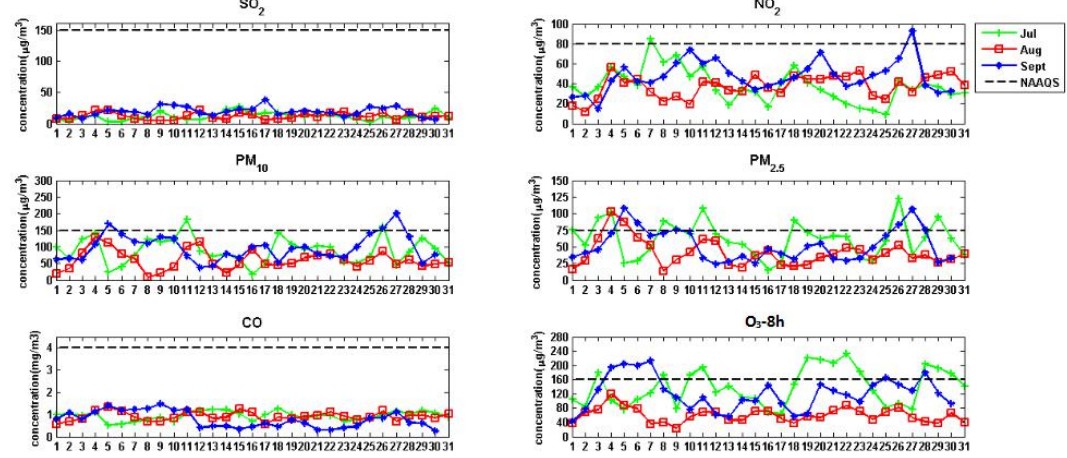


**Fig. 5.** Day-to-day variations in $SO_2$, $NO_2$, $PM_{10}$, $PM_{2.5}$, CO and $O_3$-8h at CCM station in Jul., Aug.
and Sept. 2014 (CST). Observed data in Jul., Aug. and Sept. 2014 are indicated in green, red and blue,
respectively. NAAQS are indicated in black dotted line.

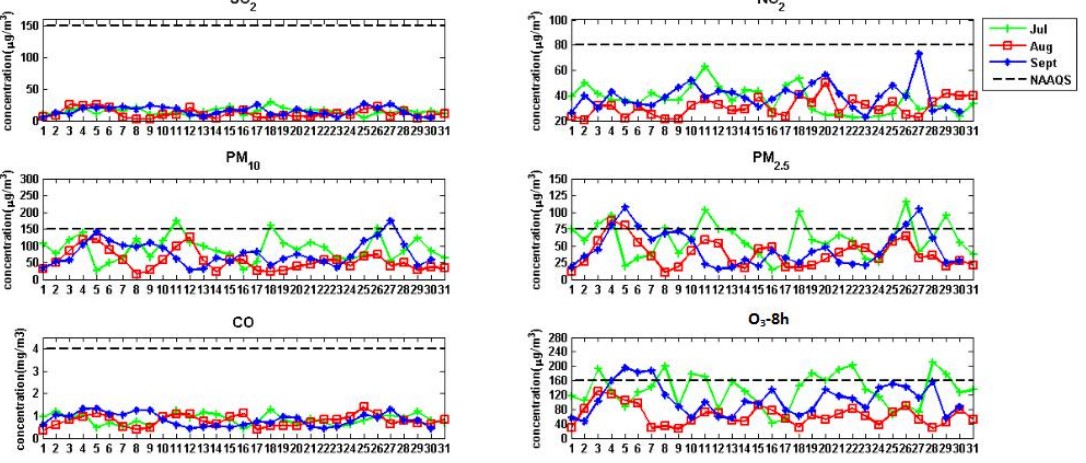


**Fig. 6.** Day-to-day variations in $SO_2$, $NO_2$, $PM_{10}$, $PM_{2.5}$, CO and $O_3$-8h at XL station in Jul., Aug. and
Sept. 2014 (CST). Observed data in Jul., Aug. and Sept. 2014 are indicated in green, red and blue,
respectively. NAAQS are indicated in black dotted line.

**Table 4**
Statistical analysis of hourly data in Jul. - Sept. 2014 at CCM station (The unit is $\mu g/m^3$ except CO
($mg/m^3$) )

| species | month | max | min | mean | median | std | Δa | Δb |
|---------|-------|-----|-----|------|--------|-----|-----|-----|
| | Jul. 2014 | 83.0 | 1.0 | 11.3 | 9.0 | 9.8 | | |
| $SO_2$ | Aug. 2014 | 72.0 | 2.0 | 11.9 | 10.0 | 7.8 | 5.1% | -37.4% |

| species | month | max | min | mean | median | std | Δa | Δb |
|---|---|---|---|---|---|---|---|---|
| | Sept. 2014 | 70.0 | 4.0 | 19.0 | 18.0 | 9.9 | | |
| | Jul.-Sept. 2014 | 83.0 | 1.0 | 14.0 | 12.0 | 9.8 | | |
| | Jul. 2014 | 161.0 | 1.0 | 37.5 | 32.0 | 28.3 | | |
| $NO_2$ | Aug. 2014 | 110.0 | 1.0 | 37.3 | 35.0 | 18.6 | -0.7% | -19.8% |
| | Sept. 2014 | 151.0 | 8.0 | 46.5 | 42.0 | 24.5 | | |
| | Jul.-Sept. 2014 | 161.0 | 1.0 | 40.2 | 37.0 | 24.4 | | |
| | Jul. 2014 | 255.0 | 6.0 | 88.5 | 88.0 | 50.7 | | |
| $PM_{10}$ | Aug. 2014 | 198.0 | 6.0 | 60.4 | 54.0 | 36.6 | -31.8% | -37.6% |
| | Sept. 2014 | 243.0 | 6.0 | 96.7 | 90.0 | 45.8 | | |
| | Jul.-Sept. 2014 | 255.0 | 6.0 | 81.7 | 76.0 | 47.4 | | |
| | Jul. 2014 | 171.0 | 1.0 | 61.5 | 58.0 | 33.9 | | |
| $PM_{2.5}$ | Aug. 2014 | 165.0 | 3.0 | 40.7 | 36.0 | 23.8 | -33.7% | -22.3% |
| | Sept. 2014 | 143.0 | 3.0 | 52.4 | 46.0 | 27.2 | | |
| | Jul.-Sept. 2014 | 171.0 | 1.0 | 51.5 | 45.0 | 29.9 | | |
| | Jul. 2014 | 2.7 | 0.2 | 0.9 | 0.9 | 0.3 | | |
| CO | Aug. 2014 | 2.2 | 0.3 | 0.9 | 0.9 | 0.3 | -1.1% | 21.1% |
| | Sept. 2014 | 2.1 | 0.1 | 0.8 | 0.7 | 0.4 | | |
| | Jul.-Sept. 2014 | 2.7 | 0.1 | 0.9 | 0.8 | 0.4 | | |
| | Jul. 2014 | 281.0 | 4.0 | 82.4 | 69.0 | 57.6 | | |
| $O_3$ | Aug. 2014 | 150.0 | 9.0 | 38.9 | 34.0 | 22.6 | -52.8% | -47.2% |
| | Sept. 2014 | 240.0 | 6.0 | 73.6 | 61.0 | 49.2 | | |
| | Jul.-Sept. 2014 | 281.0 | 4.0 | 64.7 | 51.0 | 49.3 | | |

Δa: the change percentage of species in Aug.2014 based on Jul. 2014.
Δb: the change percentage of species in Aug. 2014 based on Sept. 2014.

**Table 5**
Statistical analysis of hourly data in Jul. - Sept.2014 at XL station (The unit is $\mu g/m^3$ except CO
($mg/m^3$) )

| species | month | max | min | mean | median | std | Δa | Δb |
|---|---|---|---|---|---|---|---|---|
| | Jul. 2014 | 61.0 | 1.0 | 14.5 | 12.0 | 10.3 | | |
| $SO_2$ | Aug. 2014 | 71.0 | 1.0 | 11.4 | 8.0 | 10.4 | -21.2% | -24.6% |
| | Sept. 2014 | 75.0 | 1.0 | 15.1 | 14.0 | 10.3 | | |
| | Jul.-Sept. 2014 | 75.0 | 1.0 | 13.7 | 11.0 | 10.4 | | |
| | Jul. 2014 | 123.0 | 9.0 | 36.4 | 33.0 | 18.9 | | |
| $NO_2$ | Aug. 2014 | 95.0 | 7.0 | 30.7 | 27.0 | 15.0 | -15.8% | -21.8% |
| | Sept. 2014 | 127.0 | 11.0 | 39.2 | 36.0 | 18.7 | | |
| | Jul.-Sept. 2014 | 127.0 | 7.0 | 35.4 | 32.0 | 18.0 | | |
| | Jul. 2014 | 300.0 | 4.0 | 91.3 | 85.0 | 48.9 | | |
| $PM_{10}$ | Aug. 2014 | 196.0 | 6.0 | 55.2 | 47.0 | 35.9 | -39.6% | -28.7% |
| | Sept. 2014 | 226.0 | 9.0 | 77.3 | 70.0 | 40.3 | | |

| | | | | | | | Δa | Δb |
| --- | --- | --- | --- | --- | --- | --- | --- | --- |
| | Jul.-Sept. 2014 | 300.0 | 4.0 | 74.5 | 64.0 | 44.6 | | |
| | Jul. 2014 | 158.0 | 2.0 | 58.2 | 51.0 | 34.8 | | |
| PM$_{2.5}$ | Aug. 2014 | 157.0 | 3.0 | 38.0 | 34.0 | 24.1 | -34.6% | -17.7% |
| | Sept. 2014 | 144.0 | 3.0 | 46.2 | 38.0 | 29.0 | | |
| | Jul.-Sept. 2014 | 158.0 | 2.0 | 47.4 | 40.5 | 30.7 | | |
| | Jul. 2014 | 2.0 | 0.3 | 0.8 | 0.8 | 0.4 | | |
| CO | Aug. 2014 | 2.0 | 0.3 | 0.8 | 0.7 | 0.3 | -7.1% | -4.9% |
| | Sept. 2014 | 2.8 | 0.3 | 0.8 | 0.7 | 0.4 | | |
| | Jul.-Sept. 2014 | 2.8 | 0.3 | 0.8 | 0.7 | 0.4 | | |
| | Jul. 2014 | 238.0 | 2.0 | 78.4 | 67.0 | 55.6 | | |
| O$_3$ | Aug. 2014 | 148.0 | 2.0 | 38.7 | 32.0 | 28.3 | -50.7% | -39.9% |
| | Sept. 2014 | 226.0 | 2.0 | 64.4 | 54.0 | 46.4 | | |
| | Jul.-Sept. 2014 | 238.0 | 2.0 | 60.3 | 48.0 | 47.7 | | |

Δa: the change percentage of species in Aug.2014 based on Jul. 2014.
Δb: the change percentage of species in Aug. 2014 based on Sept. 2014.

3.2 Simulated impact of meteorological conditions
In this paper, the model configurations were the same as those set by Shu et al.
(2016), who has evaluated the model performance of WRF/CMAQ and proved the
model's reliability in simulating air quality in Nanjing.
As we know, meteorology is an important impact factor on air quality. Good
diffusion conditions can alleviate air pollution in the short term (Cermak and Knutti,
2009; Wang et al., 2009b). In this premise, if two experiments (Exp. 2 and Exp. 3) use
the same emission inventory but different weather conditions, it can be concluded that
the higher concentrations may result from poor meteorological conditions. According
to model simulation, Exp. 2 exhibited higher pollutant concentrations for all species in
most part of Nanjing as shown in Fig. 7. For SO$_2$, NO$_2$, PM$_{10}$, PM$_{2.5}$, CO, and O$_3$,
their concentrations were increased by 17.5%, 16.9%, 18.5%, 18.8%, 7.8% and 0.8%
during Aug. 2014 compared to Aug. 2013. Besides, the contributions of
meteorological conditions to primary and secondary particulate matters differed (See
Fig. 8). Secondary PM$_{10}$ (PM$_{10s}$) was raised by 21.5%, while primary PM$_{10}$ (PM$_{10p}$)
rose by 12.6% during Aug. 2014 compared to Aug. 2013. And secondary PM$_{2.5}$
(PM$_{2.5s}$) was increased by 21.5%, while primary PM$_{2.5}$ (PM$_{2.5p}$) was added by 9.5%.
Thus, the weather conditions had a slightly greater impact on secondary fine

particulate matters. Moreover, for $SO_2$, $NO_2$, $PM_{10}$, $PM_{2.5}$, CO, $PM_{10p}$, $PM_{10s}$, $PM_{2.5p}$, and $PM_{2.5s}$, there were some small decreasing areas in the northeast Nanjing (Fig. 7 and Fig. 8). In domain 4, the simulated predominant wind was northeast wind in Aug. 2014, while that was southeast wind in Aug. 2013. So, the diffusion condition of northeast Nanjing might be better in Aug. 2014 and resulted in small decrease in these areas.

The overall increasing pollutant levels in Aug. 2014 suggested that the diffusion conditions in Aug. 2014 were worse than those in Aug. 2013. Focus on the weather conditions during the YOG holding period (16-28 Aug., 2014) and the same period in 2013, the simulated hourly mean 10-m wind speed in Nanjing was larger in 16-28 Aug., 2013, and it was 1.5 m/s larger than that of 16-28 Aug., 2014 (Fig. 9). Also, the simulated 2-m temperature was higher in 16-28 Aug., 2013, and it was 2.0 K larger than that of 16-28 Aug., 2014 (Fig. 9). Besides, the simulated planetary boundary layer height (PBLH) was higher in Aug. 2013, especially in 16-28 Aug., and it was 27.5 m higher than that of 16-28 Aug., 2014 (Fig. 9). Larger wind speed and higher PBLH benefited the diffusion of air pollutants. Warming on the ground surface was conductive to the promotion of convective instability and was also good for the vertical dilution and diffusion of pollutant. Thus, the diffusion conditions in 16-28 Aug. 2013 were better than those in 16-28 Aug. 2014. Rather worse meteorological conditions in 16-28 Aug. 2014 implied that abatement controls might play a more important role in improving air quality in YOG compared with the same period in 2013. What's more, relative humidity, cloud cover and shortwave solar radiation all affect ozone chemical reaction (Katragkou et al., 2011; Pu et al., 2017). The generation and photochemical reaction of surface ozone depends on the availability of solar radiation. During the heat wave period, less relative humidity leads to less cloud cover, which could result in more net downward shortwave solar radiation and more production of $O_3$ (Pu et al., 2017). For ordinary $O_3$, its production also corresponded well to the cloud cover. As shown in Fig. 9, more relative humidity resulted in more cloud cover in northern and eastern Nanjing during 16-28 Aug., 2013, which resulted in less $O_3$ in Aug. 2013, but more $O_3$ in Aug. 2014 (Fig. 7).

340

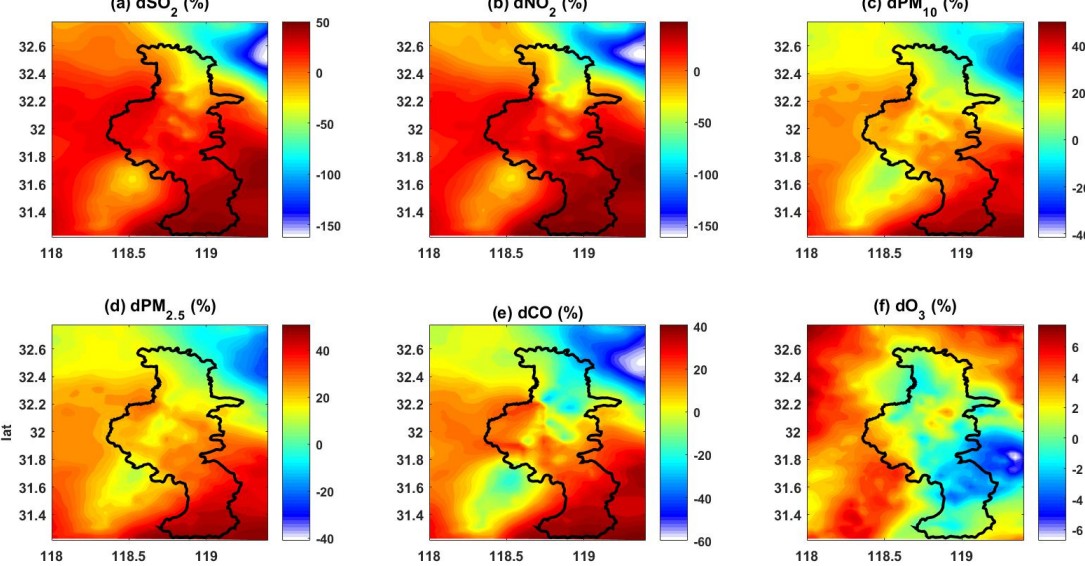

341

**Fig. 7.** Influence of meteorology on hourly mean concentrations of air pollutants in Aug. 2014

compared with Aug. 2013. (Black thick lines draw the outline of Nanjing. Picture a - f are hourly

average values of impact percentage (dspecies (%)= (Exp. 2 - Exp. 3)/Exp. 2 * 100%) of $SO_2$, $NO_2$,

$PM_{10}$, $PM_{2.5}$, CO, and $O_3$, respectively.).

346

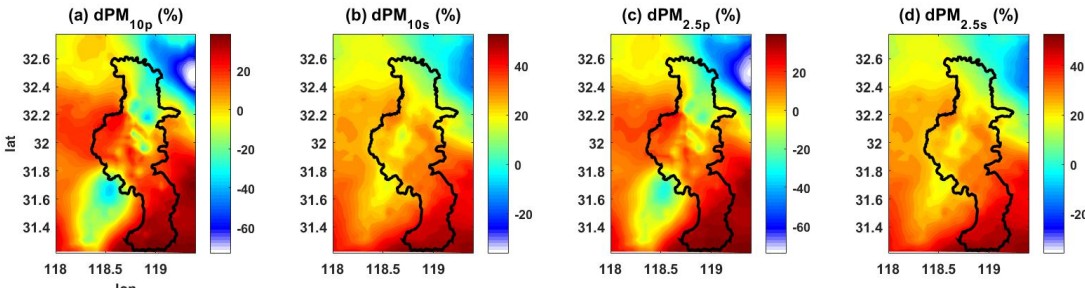

347

**Fig. 8.** Influence of meteorology on hourly mean concentrations of primary and secondary particulate

matters in Aug. 2014 compared with Aug. 2013. (Black thick lines draw the outline of Nanjing. Picture

a - d are hourly average values of impact percentage (dspecies (%)= (Exp. 2 - Exp. 3)/Exp. 2 * 100%)

of $PM_{10p}$, $PM_{10s}$, $PM_{2.5p}$, and $PM_{2.5s}$, respectively.)


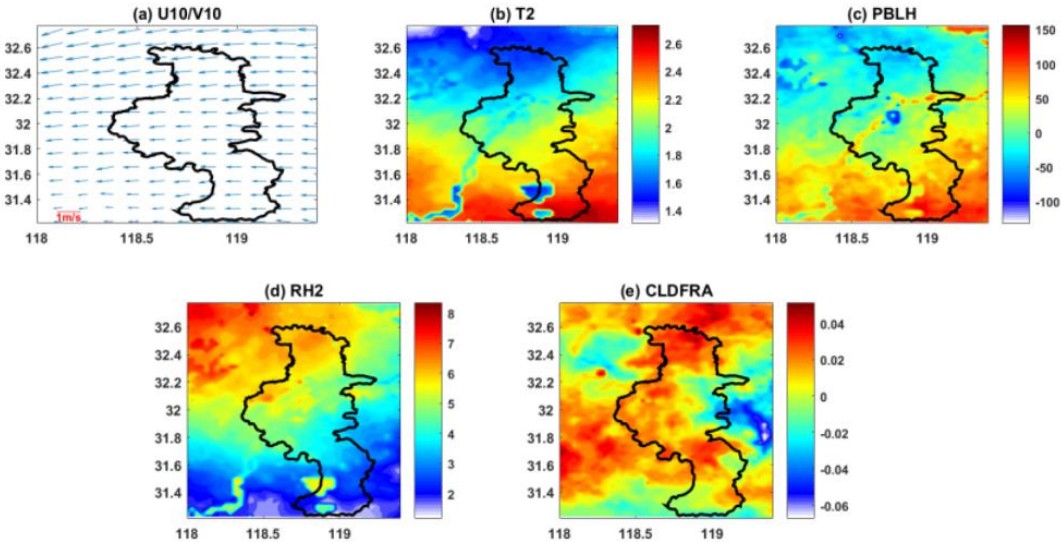


**Fig. 9.** Bias of simulated hourly mean meteorological conditions during the YOG. (Bias =
Meteorological Factors in 16-28 Aug., 2013 - Meteorological Factors in 16-28 Aug., 2014. (a) Bias of
Wind at 10m during 16-28 Aug. (unit: m/s), (b) Bias of temperature at 2m during 16-28 Aug. (unit: K),
(c) Bias of planetary boundary layer height during 16-28 Aug. (unit: m), (d) Bias of relative humidity at
2m during 16-28 Aug. (unit: %), (e) Bias of cloud fraction during 16-28 Aug.).

3.3 Simulated impact of emission reduction
As for $SO_2$, $NO_2$, $PM_{10}$, $PM_{2.5}$, and CO, the distributions of such short-lived
chemical compositions are largely affected by their sources and sinks. As seen in Fig.
10, the simulated spatial distributions of concentration changes were uneven, large
variations were found in the west of Nanjing corresponding to the downwind regions
of heavy reduction districts (See Fig. 2). Besides, impact percentages (dspecies (%) =
(Exp. 1 - Exp. 2)/ Exp. 2*100%) of species were negative except $O_3$, implying that
emission regulatory efforts were effective on the other species, but counterproductive
to $O_3$. Statistically, the concentrations of $SO_2$, $NO_2$, $PM_{10}$, $PM_{2.5}$, and CO in Nanjing
were reduced by 24.6%, 12.1%, 15.1%, 8.1% and 7.2% during Aug. 2014. As for $O_3$,
the variation was positive (1.3%), especially in the downwind area of $NO_x$ with heavy
reduction, which might due to the less titration of $O_3$ by $NO_x$ (Liu et al., 2013; Dong
et al., 2013). For primary and secondary particulate matters, the influence of emission
reduction varied dramatically. As shown in Fig. 11, $PM_{10p}$ was dropped by 39.6%,
while $PM_{10s}$ only declined by 2.9%. And $PM_{2.5p}$ was decreased by 26.2%, while
$PM_{2.5s}$ merely cut down by 2.9%. It seems that emission controls had much more
impacts on primary pollutants, especially on coarse particulate matters.

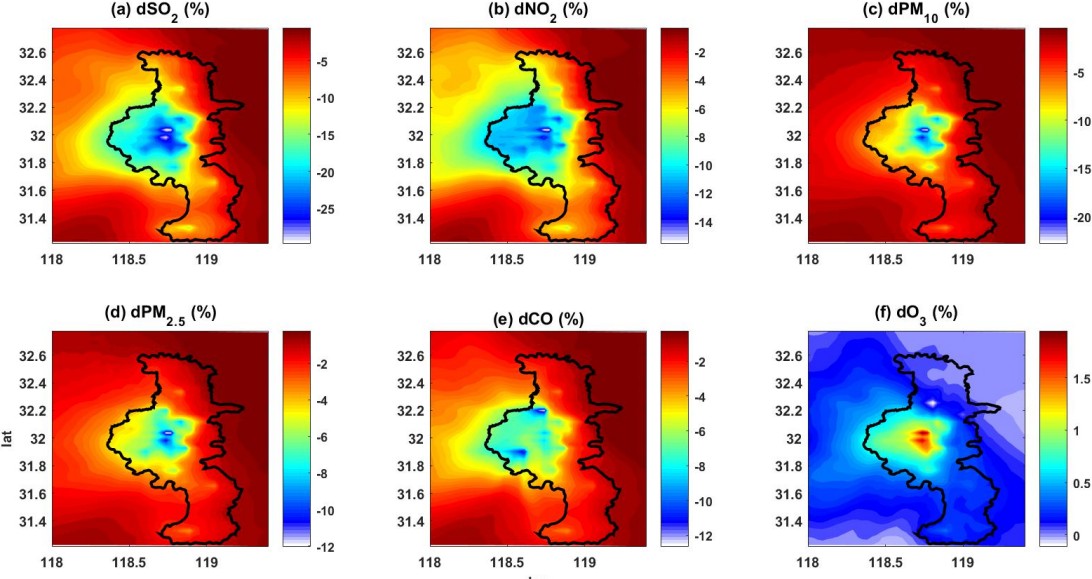


**Fig. 10.** Influence of emission reduction on hourly mean concentrations of air pollutants in Aug. 2014.
(Black thick lines draw the outline of Nanjing. Picture a - f are hourly average values of impact
percentage (dspecies (%) = (Exp. 1 - Exp. 2)/ Exp. 2*100%) of $SO_2$, $NO_2$, $PM_{10}$, $PM_{2.5}$, CO and $O_3$,
respectively.).

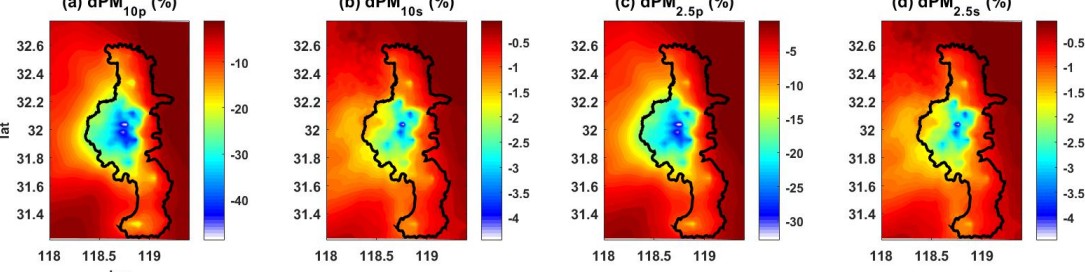


**Fig. 11.** Influence of emission reduction on hourly mean concentrations of primary and secondary
particulate matters in Aug. 2014. (Black thick lines draw the outline of Nanjing. Picture a - d are hourly
average values of impact percentage (dspecies (%) = (Exp. 1 - Exp. 2)/ Exp. 2*100%) of $PM_{10p}$, $PM_{10s}$,
$PM_{2.5p}$ and $PM_{2.5s}$, respectively.).

3.4 Comparison of simulated meteorological factors and emission reduction

391        Fig. 12 displays the simulated effect of meteorological factors and emission
reduction in Nanjing on air quality improvement during Aug. 2014. In general,
meteorological conditions played a negative role in air quality promotion in most days,
only played a positive role in a few days. And the negative effect of weather
conditions exceeded its positive effect during the whole month (See discussion in
Section 3.2). On the other hand, emission reduction contributed to the decline of $SO_2$,
$NO_2$, $PM_{10}$, $PM_{2.5}$, CO, $PM_{10p}$, $PM_{10s}$, $PM_{2.5p}$, and $PM_{2.5s}$ all the time, especially for
primary coarse particulate matters. However, reduction of $NO_x$ could cause a slight
rise of $O_3$.

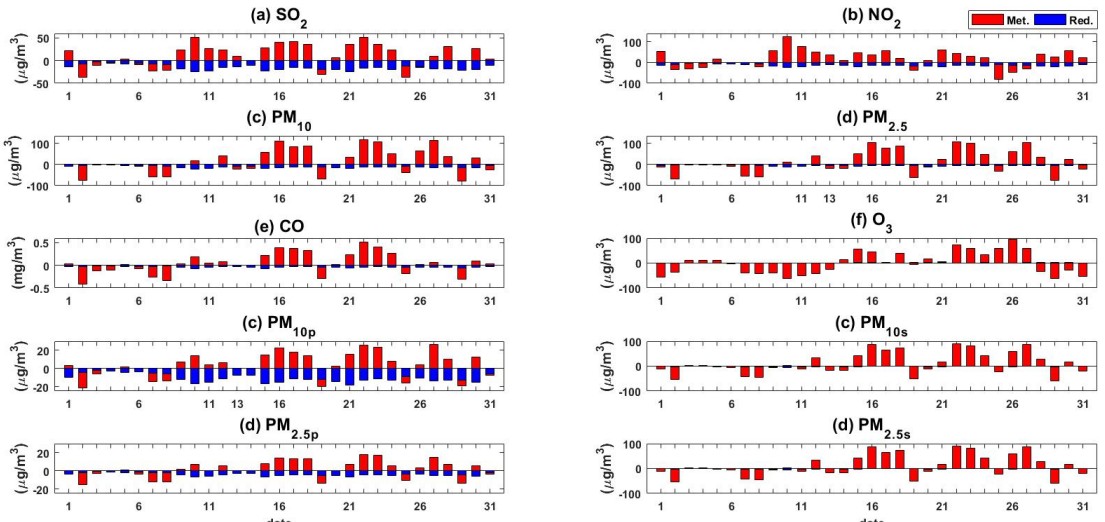


**Fig. 12.** The simulated effect of meteorology and reduction on pollutant concentrations in Nanjing
during 1-31 Aug. , 2014, Met. (Exp. 2-Exp. 3) represents the effect of meteorology, while Red. (Exp.
1-Exp. 2) represents the simulated effect of reduction.


Moreover, the opposite effects of meteorology and emission abatement on $SO_2$,
$NO_2$, $PM_{10}$, $PM_{2.5}$, CO, $PM_{10p}$, $PM_{10s}$, $PM_{2.5p}$ and $PM_{2.5s}$ during the whole month were
more apparent as statistically listed in Table 6. CCM station represents the urban
status, XL station represents the suburban status and NJ represents the whole city.
Adverse meteorology was found to raise the concentration of the six pollutants as
17.4% for $SO_2$, 15.1% for $NO_2$, 15.6% for $PM_{10}$, 14.9% for $PM_{2.5}$, 6.4% for CO and
0.9% for $O_3$ at CCM station, and 14.1% for $SO_2$, 12.4% for $NO_2$, 22.4% for $PM_{10}$,
24.5% for $PM_{2.5}$, 2.3% for CO, and 1.6% for $O_3$ at XL station. On the contrary,
emission abatement reduced their levels in most cases, especially in the urban site. It
seems that the levels of air pollutants reduced more at CCM station compared to XL
station. Emission abatement independently led to a 24.3% decrease in $SO_2$ at CCM
station, which was 5.1% higher than that at XL station. Moreover, the cutbacks of
$NO_2$, $PM_{10}$, $PM_{2.5}$ and CO were 11.7%, 13.9%, 7.5% and 7.0%, respectively at CCM
station, being larger by 1.0% to 2.0% compared with XL station. Though $O_3$ under
emission reduction scenarios resulted in a slightly rise (0.9% to 1.3%) at both sites,
the effectiveness of emission abatement was remarkable generally. As for primary and
secondary particulate matters, meteorological factors also played a negative role
during Aug. 2014, and had slightly more effect on secondary fine particles. Emission
controls seemed to cause tremendous impact on primary particulate matters,
especially for primary coarse particles. Emission abatement led to a 38.3% decrease in
$PM_{10p}$ at CCM station, a 33.2% decrease in $PM_{10p}$ at XL station, and a 39.6% decrease
in $PM_{10p}$ for the whole city. For secondary particulate matters, including sulfate,
nitrate, and ammonium salt, emission reduction played rather minor role in cutting the
pollutants, with merely a 2.4% decrease in $PM_{10s}$ and $PM_{2.5s}$ at CCM station, a 2.9%
decrease in $PM_{10s}$ and $PM_{2.5s}$ at XL station.

**Table 6**
Comparison between the simulated effect of meteorology and emission reduction at CCM and XL
station

| Species | Met. (CCM) | Red. (CCM) | Met. (XL) | Red. (XL) | Met. (NJ) | Red. (NJ) |
|---|---|---|---|---|---|---|
| $SO_2$ | 17.4% | -24.3% | 14.1% | -19.2% | 17.5% | -24.6% |
| $NO_2$ | 15.1% | -11.7% | 12.4% | -10.2% | 16.9% | -12.1% |
| $PM_{10}$ | 15.6% | -13.9% | 22.4% | -11.9% | 18.5% | -15.1% |
| $PM_{2.5}$ | 14.9% | -7.5% | 24.5% | -6.3% | 18.8% | -8.1% |
| CO | 6.4% | -7.0% | 2.3% | -5.5% | 7.8% | -7.2% |
| $O_3$ | 0.9% | 1.3% | 1.6% | 0.9% | 0.7% | 1.5% |
| $PM_{10p}$ | 13.2% | -38.3% | 5.9% | -33.2% | 12.6% | -39.6% |
| $PM_{10s}$ | 16.7% | -2.4% | 29.4% | -2.9% | 21.5% | -2.9% |
| $PM_{2.5p}$ | 8.4% | -25.8% | 4.9% | -20.1% | 9.5% | -26.2% |
| $PM_{2.5s}$ | 16.7% | -2.4% | 29.4% | -2.9% | 21.5% | -2.9% |

Met.: the change percentage of species in Exp. 2 based on Exp3, represents the effect of meteorology.
Red.: the change percentage of species in Exp. 1 based on Exp 2, represents the effect of Nanjing local
emission reduction.

The decrease of $SO_2$ might due to the limit and halt of power plants and
improvement of coal-combustion. Besides, the prohibition of heavy pollution vehicles
could contribute to the drop of $NO_2$ and CO. Also, limiting the production of
industries helped to reduce $NO_2$ and CO. The cut of particulate matters might due to
the stop of construction process and use of low ash content coal for industry. For
secondary particulate matters, controlling the emission of $SO_2$ and $NO_x$ could help to
reduce the formation of sulfate and nitrate, but no control on the emission of $NH_3$
could still result in quite a part of ammonium salt. The response of $O_3$ to emission
control could be due to its non-linearity chemistry (Fu et al., 2012), reducing $NO_2$
pollution may have side-effect by increasing $O_3$ because of the titration effect. On the
whole, the meteorological factors and emission reduction during the YOG had
opposite effects on $SO_2$, $NO_2$, $PM_{10}$, $PM_{2.5}$, and CO, and emission reduction played a
very important role in the air quality improvement.

**4 Summary and conclusions**
The air quality during the 2nd YOG was superior according to the current
NAAQS. Both observation and modeling confirmed that stringent emission reductions
were effective to ambient air quality promotion during the Youth Olympic Games,
especially to $SO_2$, $NO_2$, $PM_{10}$, $PM_{2.5}$ and CO. Compared to Aug. 2013, the observed
concentrations in Aug. 2014 were dropped by 64.7% for $SO_2$, 29.8% for $PM_{10}$, 9.8%
for $PM_{2.5}$, 8.9% for CO and 31.7% for $O_3$ at CCM station, while 50.0% for $SO_2$,
18.6% for $NO_2$, 32.8% for $PM_{10}$, 4.1% for $PM_{2.5}$, and 31.7% for $O_3$ at XL station. The
simulated impact percentage of emission reductions were -24.6%, -12.1%, -15.1%,
-8.1% and -7.2% for $SO_2$, $NO_2$, $PM_{10}$, $PM_{2.5}$, and CO, respectively.
The meteorological conditions in the holding time of the YOG were inferior to
those of the same period in 2013, with lower temperature and weaker winds. Model
simulations show that less favorable weather conditions caused higher concentrations
for all species, including $O_3$ which was increased due to less cloud cover. Besides,
meteorological conditions had slightly more effect on secondary fine particular
matters compared to primary fine particular matters. Emission reduction could cut
down the levels of $SO_2$, $NO_2$, CO and particular matters, especially for primary coarse
particles. Thus, emission reduction control is the very important factor for the air
quality improvement during the YOG.
In general, better air quality during the YOG benefit a lot from emission
reduction, which has set up a good example in air protection for important social
events. However, the enhanced concentrations of air pollutants after the YOG (in Sept.
2014) suggest that short-term emission control can only ease air pollution effectively
and temporarily. Long-term control policies are necessary to ensure pleasant future air
quality.

**Acknowledgements**
This work was supported by the National Natural Science Foundation of China
(91544230, 41575145, 41621005), the National Key Research Development Program
of China (2016YFC0203303, 2016YFC0208504, 2014CB441203), the National
Special Fund for the Weather Industry (GYHY201206011) and the National Special
Fund for the Environmental Protection Industry (GYHY201409008) . We are grateful
to Prof. Yu Zhao from School of Environment of Nanjing University for supply the
emission data of Nanjing. The contents of this paper are solely the responsibility of
the authors and do not necessarily represent the official views of sponsors.

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
