# Peer review of "Impacts of emission reduction and meteorological conditions on air"

_Atmospheric Chemistry and Physics, 2017_

## Referee Comment (RC1) · Anonymous Referee #1 · 10 Mar 2017

This manuscript described a study for the emission control scenario during the 2nd Youth Olympic Games in Nanjing using surface measurements and WRF/CMAQ model. This manuscript's English need improvement. It listed both model and measurement results, but it is not easy to track which is about observation and which is about the model. I suggest add something to make it clear. For instance, the title of section 3.1 can be "Observed air quality during YOG", and the section 3.2 changes to be "simulated impact of meteorological conditions". Another issue is that the discussions for the measurements and model are totally separated, and the modeled impact of NOx emission reduction on O3 et al is not supported by the observation. Obviously

the model or emission inventory has some biases, which should be addressed. Another issue is that this study did not discuss anything about emission and pollutant concentrations in surrounding areas, which sometimes can affect your results. Page 1, line 27, "However, simulation" better to be "However, the model simulation" Page 1, line 28, "and raised SO2" better to be "and could increase" Page 2, line 48-49, "Preparatory work were carried out since 1 July, 2014" better to be "The preparation started from July 1, 2014" Page 2, line 54-59. Please consider to split that long sentence to several sentences as it has grammar errors. Page 4, line 137. "Exp.3 had the same inventory as Exp.2 but the weather" better to be "Exp.3 had the same inventory as Exp.2 but used the weather" Page 4, line 141. "meteorology on contaminants" better to be " meteorology on air quality" Page 6, line 132-141. This manuscript should show a map of the emission reduction for Exp1 –Exp2, instead of just modeled concentration changes. Page 8, line 184, "most species had a good reflection", What does it mean? Page 8, line 186-194. Please re-write to make it easy to understand. Figure 6, 7 and the corresponding discussion in section 3.2. Are those comparisons for monthly averaged value, such as 10m wind, PBL heights? If so, please state it. Page 13, line 257-259. The O3 increase should be due to the NOx emission reduction -> less titration. Page 14, table 6. Why the modeled impact of the emission reduction on NO2, O3, PM10, PM2.5 diff significantly from the observations? You may discuss it.

---

## Referee Comment (RC2) · Anonymous Referee #2 · 16 Mar 2017

This study investigated the variation of and the relative contributions of meteorology and emission reduction to air quality during the 2014 Youth Olympic Games in Nanjing through observation analysis and WRF-CMAQ simulation. It states that under unfavorable meteorological conditions, emission reduction is a dominant factor to improve air quality, which is successful in controlling air pollution during the event in Nanjing. This study could be useful to the understanding of haze formation mechanism in east China related to meteorology and emission variation, however, this version of manuscript lacks in-depth analysis of pollutant evolution and some interpretation is not sound and insufficient, so I recommend a major revision of this manuscript.

[Figure]

Specific comments:

1. line 112-113, 'the 9 state controlling air sampling sites in Nanjing were chosen to represent the whole Nanjing'. Looking at fig 1b, I found 9 stations almost concentrate in the urban area, which is small compared with the whole Nanjing, so I doubt the 9 sites can represent the whole Nanjing, and it's better to collect some observations at rural sites of Nanjing for model validation.

2. Line 149-160 presents the comparison between August 2014 and August 2013 and states ' emission reductions did help the alleviation of air pollution......', you didn't look at and discuss the difference in meteorological conditions between the two years, how can you rule out the potential influence of meteorology, so please add meteorology comparison here. Also, there are evident emission reductions during August 2014, with 22.1% for $SO_2$, 12.5% for $NO_x$, and 21.4% for $PM_{2.5}$, why the decrease in $PM_{2.5}$ concentration at CCM is just 9.8%, how about the proportion and relative changes of primary and secondary $PM_{2.5}$?

3. Line 182-191, when comparing simulations in August 2014 with that in July and September 2014, you try to say ' the pollutant concentrations declined with emission control, but rebounded after releasing control', however, the simulated $SO_2$ concentration in August is larger than that in July (5.1%), whereas $NO_2$ (19.8%) and CO (21.1%) in August are larger than in September, how do you explain the larger $SO_2$, $NO_2$ and CO concentrations in August although strict emission abatement is implemented than those in July and September with no emission reduction?

4. line 227-228, 'Consequently, Exp.2 resulted in higher pollutant concentrations for all species as shown in Fig.7', this is not true, although the domain averages of pollutant values increase from Exp3 to Exp2, it is apparent that the spatial distribution did not show a consistent increase in the domain, such as the large decreases in all components but $O_3$ to the northeast, and the decreases in $SO_2$, $NO_2$, CO, $O_3$ in portions of Nanjing, so the meteorological condition in August 2014 did not necessarily lead to

increases in pollutant levels, so I suggest more discussion on the different responses to meteorology in the domain with analysis of meteorological variable changes.

5. section 3.3, similar problems in this section, when emission reduction lead to apparent decreases in concentrations of all pollutants except O3, how do you explain the apparent increases in the southern parts of Nanjing (Fig. 8)? are there some feedbacks among aerosols, radiation (photolysis), cloud and consequent effects on chemical processes, please elaborate on mechanisms behind these changes instead of just presenting model results.

6. Regarding Fig. 9, please explain how the meteorological change lead to day-to-day variations (either increase or decrease) of pollutant concentration.

7. Some tables like Table 4 and 5 can be removed because this manuscript is not a data report.

8. Please describe clearly the spatial and time scales of the presented data or model results and the comparison between cases throughout the manuscript, such as line 266 ' Fig.9 displays the effect of meteorological factors and emission reduction', please write clearly the numerical experiments, the time period and which domain for average etc.

9. The English in this manuscript should be carefully checked and much improved by correcting grammatical errors and rewording sentences, some of them are misleading and ambiguous.

---

## Referee Comment (RC3) · Anonymous Referee #3 · 28 Mar 2017

**General Comments**

This paper tried to evaluate the impacts of emission reduction and meteorological conditions on the air quality improvement during an air pollution control period-YOG of Nanjing. Accurate quantification of the influence of emission reduction and meteorological conditions is important to evaluate the air pollution control measures. This paper used both observation data and modeling results to address this issue. However, this manuscript has major writing and structure problem. 1 The validation of model simulation and uncertainty analysis is essential and required but lack in the manuscript. 2 The paper lacks in-depth discussions of the observation data and model results. Some conclusions are too arbitrary and lack sufficient evidence to back the interpretations of the results (see detail comments below). 3 The literature review in the introduction section needs improvement. 4 The quality of English needs substantial improving. I believe that the paper needs substantial revisions before considering to be published at ACP.

**Detail Comments**

1 Line 22-26 This sentence here is not rigorous. What concentration? Hourly average? Daily average? From what data? Observation data at which site? You'd better give the standard deviations of the data.

2 The introduction section should be rewritten and reorganized. The references cited in the introduction section should be more targeted and well selected. Take Line 78-82 for example, the references cited here have nothing to do with the topic of the paper. Line 60-82, too many references are cited without summary and in-depth understanding.

3 Section 2.1, Line 97-103, the data description here is too simple and lack of important information. Why CCM and XL station are chosen for study? Can they represent the whole study

area of the modeling or Nanjing? What instruments are used for observation? How about the quality of the data and the uncertainty of the measurements?

3 Section 2.2, Fig. 1 is hard to read. The authors stated that the 9 stations were chosen for representing the whole Nanjing city. But all of the 9 stations located at the center part of the city. I doubt can they represent the whole city? Moreover, what is the purpose of these sites? For model validation? Please give the results of model validation.

4 Section 2.3 The description here is quite ambiguous. Which year of the emission inventory is used for simulation? How do the authors make the emission inventory after reduction? How to determine the reduction ratio? Based on the control measures? Is there any hypothesis here? If there is hypothesis? What is the uncertainty? Please state the experimental process in detail.

5 Section 3.1 The title of this section is inconsistent with the content. Why CCM and XL station are chosen for study? Can they represent the whole study area of the modeling or Nanjing (same as detail comments NO. 3)? The data analysis in this section should be more rigorous and more in-depth. Line 147-148 How to get the reduction percentage? Calculate from observation data or other ways? Line 154-156 Why the authors avoid discussion of $NO_2$ at CCM and CO at XL? Line 157-158 The discussion here is inaccurate. The deviation of $PM_{10}$ and $PM_{2.5}$ is larger in 2014. Line 158-160 How to get this conclusion from the analysis above? Line 182-199 Similar problems as above. Line 190-191 The change percentage of $NO_2$ listed here is 19.8 %, but in Table 4 is -19.8%, please check the correctness and consistency of your results. In line 193-194, the authors said that "the pollutant concentrations declined with emission control, but rebounded after releasing control". How to explain the higher simulated concentrations of $SO_2$ and CO during August with strictly control measures? The authors listed too many tables in this section without in-depth analysis and solid discussions.

6 Line 221-232, The authors should avoid ambiguous discussion. The word such as "lower temperature and weaker winds", "rather worse meteorological conditions" is quite obscure to readers. Line 227, The authors stated "…… which was consistent with the observations", could you give more detailed comparison results of model and observations? How about the accuracy

of the simulated meteorological parameters? Fig. 6, What do "data1" and "data2" stand for? Fig. 7 How to explain the spatial distributions of the impact percentage? For CO and $O_3$, the simulated concentrations of Exp. 2 are lower than those of Exp. 3, especially for the north part of Nanjing city.

8 Section 3.3, Line 247-248, the statement here is ambiguous. 9.2% and 38.1% is from model results or others? 9.2% to 38.1% is a fuzzy range. Line 249-250, what do you mean? What is the definition of short-lived chemical composition? Line 250-251 How to explain the uneven distribution of the impact percentage? Line 256-257 The reduction ratios here are compared to what period? The authors should give more exact time during the discussion.

9 Section 3.4 Why do you choose 16[th] Aug. to 28[th] Aug. not the whole month of August as the study time here? Line 270-271 How can you make the conclusion here? From Fig. 9, it seems that the influence of meteorological conditions is more important for the air quality of Nanjing. Line 278-291 The authors focus on discussing difference of emission reduction influence at two sites. However, 0.9 %, 1.1 % etc. is quite small change. What is the result when considering the uncertainty of the model simulations? Line 299-308 The discussions here lack of evidence.

**Technical Comments**

1 The authors should refer to "the guidelines for authors" of ACP to prepare the manuscript.

2 Abbreviations should be given for the first time. Such as "CST" etc.

3 The date format need to be uniform.

4 Spaces must be included between number and unit.

5 Fig. 9 The legend makers "Met" and "Red" here are easy to lead misunderstanding. You'd better use "Met." and "Red.".

6 The reference format should be uniform. Too many references in Chinese are cited.

7 The English of this manuscript needs substantial improvement.

---

## Author Comment (AC1) · 8 Jun 2017

Thank you very much for reviewing the manuscript and providing us the constructive comments and suggestions on our study. We have learned a lot from your advice and revised the manuscript, which we hope meet with approval. And point-by-point responses are listed as below:

Responses to the reviewer's comments:

Comment 1: This manuscript described a study for the emission control scenario

during the 2nd Youth Olympic Games in Nanjing using surface measurements and WRF/CMAQ model. This manuscript's English need improvement. It listed both model and measurement results, but it is not easy to track which is about observation and which is about the model. I suggest add something to make it clear. For instance, the title of section 3.1 can be "Observed air quality during YOG", and the section 3.2 changes to be "simulated impact of meteorological conditions". Response: Thank you for your advice. We have followed your advice and modified the manuscript to make observation part and modeling part more clearly. The title of section 3.1 can be "Observed air quality during YOG", and the section 3.2 changes to be "simulated impact of meteorological conditions".

Comment 2: Another issue is that the discussions for the measurements and model are totally separated, and the modeled impact of NOx emission reduction on O3 et al is not supported by the observation. Obviously the model or emission inventory has some biases, which should be addressed. Response: Thank you for your advice. The measurements and model are separated because we want to investigate the variations of air pollutants from different point of view, one is the real reduction of air pollution from observation, the other is model reduction from meteorology and emission. The observational decrease of O3 may due to the meteorological conditions. In Aug. 2014, there were more overcast days, and the reduction in solar radiation could restrain the production of O3. However, in the model simulation, underestimation of cloud cover could result in more solar radiation, which was conductive to the promotion of O3. Besides, reduction of NOx could result in less titration of O3 by NOx, which also lead to higher simulation O3. Thus, the observational O3 variation and the model simulation O3 variation are different, which was discussed in section 3.3.

Comment 3: Another issue is that this study did not discuss anything about emission and pollutant concentrations in surrounding areas, which sometimes can affect your results. Response: Thank you for your comment. During the YOG holding month (Aug. 2014), though the surrounding area of Nanjing had taken part in the pollution emission

control, their magnitudes of emission reduction were very small. And the emission reduction were mainly concentrated in the holding city, Nanjing, so we focus on the local emission and pollutant concentrations.

Comment 4: Page 1, line 27, "However, simulation" better to be "However, the model simulation" Response: We are grateful for your suggestion. We have followed your advice and modified the manuscript (Line 27, Page1).

Comment 5: Page 1, line 28, "and raised SO2" better to be "and could increase" Response: We have modified it according to the comment (Line 29, Page1).

Comment 6: Page 2, line 48-49, "Preparatory work were carried out since 1 Jul., 2014" better to be "The preparation started from Jul. 1, 2014". Response: We have changed it according to the suggestion (Line 48-49, Page 2).

Comment 7: Page 2, line 54-59. Please consider to split that long sentence to several sentences as it has grammar errors. Response: Thank you for your advice. We have followed your suggestions and modify the manuscript. And the sentences (Line 52-57, Page 2) have been rephrased as follows: Some local petrochemical, chemical and steel industries were forced to limit or halt their production. Coal-combustion enterprises were required to use high-quality coals with low sulfur content and ash content. And heavy pollution vehicles called "yellow label buses" were prohibited in Nanjing during 10-28 Aug.. Oil loading and unloading operations were strictly controlled. Construction process was forced to stop.

Comment 8: Page 4, line 137. "Exp.3 had the same inventory as Exp.2 but the weather" better to be "Exp.3 had the same inventory as Exp.2 but used the weather" Response: We have changed it according to the suggestion (Line 203, Page 8).

Comment 9: Page 4, line 141. "meteorology on contaminants" better to be "meteorology on air quality" . Response: We have changed it according to the suggestion (Line 206, Page 8).

Comment 10: This manuscript should show a map of the emission reduction for Exp1
- Exp2, instead of just modeled concentration changes. Response: Thank you for
your advice. We have add some introduction of the emission inventory used in model
simulation in Section 2.3 Emissions and simulation scenarios and offer the map (See
Fig.2) of the emission reduction for Exp.1 - Exp.2.

Comment 11: Page 8, line 184, "most species had a good reflection", What does
it mean? Response: Thank you for your comment. We're sorry about the confusing
sentence and have rewritten it. It means the concentrations of most species decreased
obviously in Aug. 2014 compared with those in Jul. 2014 and Sept. 2014 (Line 249,
Page 11).

Comment 12: Page 8, line 186-194. Please re-write to make it easy to understand.
Response: Thank you for your suggestion. We have rewritten the sentences (Line
251-260, Page 11).

Comment 13: Figure 6, 7 and the corresponding discussion in section 3.2. Are those
comparisons for monthly averaged value, such as 10m wind, PBL heights? If so, please
state it. Response: Thank you for your comment. We have completed the captions
of the figure (Line 325-328). And Figure 7 shows hourly average values of impact
percentage (dspecies (%)= (Exp.2 - Exp.3)/Exp.2 * 100%) of SO2, NO2, PM10, PM2.5,
CO, and O3, respectively. To better display the bias of meteorological parameters,
Figure 6 was replaced by Figure 8 in the revised manuscript (Line 329-334, Page 16),
they're hourly average values.

Comment 14: Page 13, line 257-259. The O3 increase should be due to the NOx
emission reduction -> less titration.ãĂĂ Response: Thank you for your comment. We
have corrected it (Line 340-342, Page 16) as follows: "As for O3, the variation was
positive, especially in the downwind area of NOx heavy reduction region, which might
due to the less titration of O3 by NOx"

Comment 15: Page 14, table 6. Why the modeled impact of the emission reduction

on NO2, O3, PM10, PM2.5 diff significantly from the observations? You may discuss it. Response: Thank you for your suggestion. As discussed in Section 2.3 (Line 197-206), the simulation scenarios are reasonable. And the dynamic parameterization in WRF as well as the physical and chemical schemes of CMAQ applied in this research were the same as those in the research of Shu et al.(Shu is a member of our research group). The model performance has been validated by Shu et al., and they proved that WRF/CMAQ is reliable as shown in the uploaded Fig.1 and Fig.2.

Several factors contribute to the bias of NO2, O3, PM10, PM2.5 between simulation and observation. Firstly, a overall underestimation of emission reduction might result in less variations of NO2, PM10, and PM2.5 caused by emission abatement. Secondly, the observational O3 decreased during the emission control month (Aug. 2014) while the simulation O3 increased slightly under emission control, which can be affected by the cloud simulation and the modeling chemical mechanism. During Aug. 2014, there were more overcast days, which may cause less solar radiation and was adverse for the promotion of O3. However, the underestimation of modeling cloud cover could lead to higher simulation O3. Besides, reduction of NOx could result in less titration of O3 by NOx, and overestimation of this chemical mechanism could also lead to higher simulation O3. What's more, the aim of Table 6 (Page 18) is to compare the simulated effect of meteorological conditions and emission reduction other than comparing the simulation with the observation. It want to indicate that the adverse meteorological conditions in Aug. 2014 could cause the increase of pollution concentrations while emission reduction could help to cut down the pollutants' (SO2, NO2, CO, PM10, and PM2.5) level during Aug. 2014.
* * *
**Table 3.** Comparisons between the simulations and the observations at Shanghai, Nanjing and Hangzhou stations during 4–15 August 2013.

| Site[a] | Vars[b] | Mean | | $R$[e] | NMB[f] | RMSE[g] |
|---|---|---|---|---|---|---|
| | | OBS[c] | SIM[d] | | | |
| SH | $T_2$ (°C) | 33.27 | 31.38 | 0.91 | −5.68 % | 4.15 |
| | $RH_2$ (%) | 57.91 | 65.23 | 0.85 | 12.64 % | 19.3 |
| | $Wspd_{10}$ (m s$^{-1}$) | 4.59 | 4.66 | 0.77 | 1.53 % | 2.18 |
| | $Wdir_{10}$ (°) | 176.34 | 182.57 | 0.63 | 3.53 % | 41.44 |
| | $O_3$ (ppb) | 87.77 | 82.5 | 0.81 | −6.00 % | 38.79 |
| | $NO_2$ (ppb) | 29.01 | 38.25 | 0.54 | 31.85 % | 28.95 |
| NJ | $T_2$ (°C) | 32.95 | 30.98 | 0.84 | −5.98 % | 2.91 |
| | $RH_2$ (%) | 63.28 | 66.14 | 0.83 | 4.52 % | 9.41 |
| | $Wspd_{10}$ (m s$^{-1}$) | 3.21 | 3.4 | 0.74 | 5.92 % | 2.41 |
| | $Wdir_{10}$ (°) | 197.68 | 194.58 | 0.57 | −1.57 % | 71.19 |
| | $O_3$ (ppb) | 69.7 | 78.15 | 0.81 | 12.12 % | 36.8 |
| | $NO_2$ (ppb) | 41.44 | 40.09 | 0.61 | −3.26 % | 22.4 |
| HZ | $T_2$ (°C) | 33.25 | 31.08 | 0.8 | −6.53 % | 3.09 |
| | $RH_2$ (%) | 52.76 | 61.39 | 0.78 | 16.36 % | 13.96 |
| | $Wspd_{10}$ (m s$^{-1}$) | 3.04 | 3.32 | 0.75 | 9.21 % | 2.39 |
| | $Wdir_{10}$ (°) | 186.45 | 186.2 | 0.58 | −0.13 % | 69.44 |
| | $O_3$ (ppb) | 76.57 | 84.51 | 0.83 | 10.37 % | 33.95 |
| | $NO_2$ (ppb) | 31.06 | 27.21 | 0.66 | −12.40 % | 16.86 |

[a] Site indicates the city where the observation sites locate, including Shanghai (SH), Nanjing (NJ) and Hangzhou (HZ). [b] Vars indicates the variables under validation, including 2 m air temperature ($T_2$), 2 m relative humidity ($RH_2$), 10 m wind speed ($Wspd_{10}$), 10 m wind direction ($Wdir_{10}$), ozone ($O_3$) and nitrogen dioxide ($NO_2$). The words between the parentheses behind variables indicate the unit. [c] OBS indicates the observation data. [d] SIM indicates the simulation results from WRF/CMAQ. [e] $R$ indicates the correlation coefficients, with statistically significant at 95 % confident level. [f] NMB indicates the normalized mean bias. [g] RMSE indicates the root-mean-square error.

**Fig. 1.** Figures for Comment 15, Fig1. model performance

[Figure]

**Figure 6.** Hourly variations of the observed and the simulated $O_3$ concentrations in Shanghai, Nanjing and Hangzhou during 4 to 15 August 2013. The red solid lines show the modeling results, the black dot lines give the observations, and the solid gray lines represent the national standard for the hourly $O_3$ concentration, which is $200 \, \mu g \, m^{-3}$.

**Fig. 2.** Figures for Comment 15, Fig2. model performance

---

## Author Comment (AC2) · 8 Jun 2017

Thank you very much for reviewing the manuscript and providing us the constructive comments and suggestions on our study. We have learned a lot from your advice and revised the manuscript, which we hope meet with approval. And point-by-point responses are listed as below:

Responses to the reviewer's comments:

Comment 1: line 112-113, 'the 9 state controlling air sampling sites in Nanjing were

chosen to represent the whole Nanjing'. Looking at fig 1b, I found 9 stations almost concentrate in the urban area, which is small compared with the whole Nanjing, so I doubt the 9 sites can represent the whole Nanjing, and it's better to collect some observations at rural sites of Nanjing for model validation.

Response: Thank you for your comment. Your suggestion is reasonable, but the limitation is that there are only 9 state controlling air sampling sites in Nanjing as shown in the paper. Among them, XL site is regarded as a suburban site, while CCM site is regarded as an urban site. Besides, the Nanjing Municipal Environmental Protection Bureau takes the 9 state controlling air sampling sites to represent the whole Nanjing and issues Nanjing Air Quality Daily Report. So we think it may be better to follow the local EPA. The details about choosing the sites have been added in Section 2.2 (Line 151-155, Page 5).

Comment 2: Line 149-160 presents the comparison between Aug. 2014 and Aug. 2013 and states' emission reductions did help the alleviation of air pollution......', you didn't look at and discuss the difference in meteorological conditions between the two years, how can you rule out the potential influence of meteorology, so please add meteorology comparison here. Also, there are evident emission reductions during Aug. 2014, with 22.1% for SO2, 12.5% for NOx, and 21.4% for PM2.5, why the decrease in PM2.5 concentration at CCM is just 9.8%, how about the proportion and relative changes of primary and secondary PM2.5?

Response: Thank you for your suggestion. We have reorganized this part (Line 222-228). And we have followed your advice and comparison of meteorological conditions is in Section 3.2. The emission reduction percentages are the mean of the whole city, the details of emission reduction are added in Section 2.3 Line 173-195. However, the distributions of emission reduction are not uniform since the intensities of emission reduction are different from various trades. Also, the relationship between pollutant source and pollutant concentration is not linear. Thus, the decrease in PM2.5 concentration at urban site CCM is not very big. The PM2.5 observational data is total PM2.5,

so we can't distinguish the proportion and relative changes of primary and secondary PM2.5.

Comment 3: Line 182-191, when comparing simulations in Aug. 2014 with that in Jul. and Sept. 2014, you try to say the pollutant concentrations declined with emission control, but rebounded after releasing control, however, the simulated SO2 concentration in Aug. is larger than that in Jul. (5.1%), whereas NO2 (19.8%) and CO (21.1%) in Aug. are larger than in Sept., how do you explain the larger SO2, NO2 an CO concentrations in Aug. although strict emission abatement is implemented than those in Jul. and Sept. with no emission reduction?

Response: Thank you for your comment. Firstly, this paragraph compares the observational data other than simulations in Aug. 2014 with that in Jul. and Sept. 2014. Secondly, we're sorry that there is a mistake in Line 190 (old manuscript): "the change percentage of species (SO2, NO2, PM10, PM2.5, CO and O3) was -37.4%, 19.8%, -37.6%, -22.3%, 21.1%, and -47.2%, respectively at CCM station (Table 4)", "19.8%" should be "-19.8%", and we have corrected it (Line 259-260). Thirdly, Table 4 and Table 5 show the observational pollutants variations other than simulated pollutants variations in Jul., Aug. and Sept. at CCM station and XL station. We can see that at XL station (a suburban station), the concentrations of all species (SO2, NO2, PM10, PM2.5, CO, and O3) in Aug. are the lowest compared to those in Jul. and Sept.. Besides, at CCM station (a urban station), the concentrations of most species (NO2, PM10, PM2.5, and O3) are the lowest compared to those in Jul. and Sept.. These show a pollutant concentration decline trend after emission control and a rebound trend after releasing control. Besides, at CCM station, the observational SO2 concentration in Aug. is larger than that in Jul. (5.1%), whereas CO (21.1%) in Aug. are larger than in Sept., which could be caused by many factors, such as traffic and other unpredictable emissions around the site. As for traffic control, only the heavy pollution vehicles called "yellow label buses" were prohibited in Nanjing during 10-28 Aug.. To meet the traffic demand of numerous tourists, athletes, and freightage, there could be more traffic pollution and

raised the level of SO2, NOx and CO. Besides, NOx was mainly emitted from power plants, so the overall NOx was the lowest during the emission control month. However, they don't bother the overall variation trend of the six species.

Comment 4: line 227-228, 'Consequently, Exp.2 resulted in higher pollutant concentrations for all species as shown in Fig.7', this is not true, although the domain averages of pollutant values increase from Exp3 to Exp2, it is apparent that the spatial distribution did not show a consistent increase in the domain, such as the large decreases in all components but O3 to the northeast, and the decreases in SO2, NO2, CO, O3 in portions of Nanjing, so the meteorological condition in Aug. 2014 did not necessarily lead to increases in pollutant levels, so I suggest more discussion on the different responses to meteorology in the domain with analysis of meteorological variable changes.

Response: Thank you for your comment. This paper tries to discuss the overall impact of meteorological conditions. Based on Fig.7, statistics show that meteorology in Aug. 2014 led to total increases in pollutant levels. Line 301-302 offer the details : "For SO2, NO2, PM10, PM2.5, CO, and O3, their concentrations were increased by 17.5%, 16.9%, 19.0%, 19.5%, 7.8% and 0.8%". Factors such as topography could affect locally, and may cause discontinuous increases in Fig.7 , but it did not affect the overall increase trend. So, partial decrease is not that important. The analysis of meteorological variable changes was in Line 303-314.

Comment 5: section 3.3, similar problems in this section, when emission reduction lead to apparent decreases in concentrations of all pollutants except O3, how do you explain the apparent increases in the southern parts of Nanjing (Fig. 8)? are there some feedbacks among aerosols, radiation (photolysis), cloud and consequent effects on chemical processes, please elaborate on mechanisms behind these changes instead of just presenting model results.

Response: Thank you for your comment. The mechanisms about meteorology have been discussed in Section 3.2. In Section 3.3, the apparent increases of (SO2, NO2,

[Figure]

PM10, PM2.5 and CO) in the southern parts of Nanjing seems unreasonable. To find out the reason, we carefully checked our data processing, simulation scenarios, and emission inventory. We found that the problem was not caused by meteorology but the emission inventory. The emission inventory used in Exp.1 (under emission control) had some problem with some points larger than those in the emission inventory before emission control. The emission under control should not exceed the emission before control. We are sorry about that. We have corrected the emission inventory (under emission control), redone the model simulation of Exp.1 and reprocessed the data. And the corrected figure (See Section 3.3, Fig.9) as shown below don't have increases (SO2, NO2, PM10, PM2.5 and CO) in the southern of Nanjing. Besides, emission reduction led to completely decrease (SO2, NO2, PM10, PM2.5 and CO) in the whole city, and increase of O3 in Nanjing. The drop of O3 was due to the reducing NO2 and less titration impacts.

Comment 6: Regarding Fig.9, please explain how the meteorological change lead to day-to-day variations (either increase or decrease) of pollutant concentration.

Response: Thank you for your comment. The old Fig.9 is the current Fig.10 in section 3.4, it aims to compare the simulated effect of meteorology and emission reduction from day to day during the YOG other than to explain how meteorological change lead to day-to-day variations. From Fig.10 , we can see that emission control caused decreases of pollutant (SO2, NO2, PM10ïijŇPM2.5, and CO) concentration while meteorology caused increases of pollutant (SO2, NO2, PM10, PM2.5, CO and O3) concentration in most of the time.

Comment 7: Some tables like Table 4 and 5 can be removed because this manuscript is not a data report.

Response: Thank you for your advice. Table 4 and Table 5 are statistical analysis of observational data, we think they're important and retain them may be better.

Comment 8: Please describe clearly the spatial and time scales of the presented data

or model results and the comparison between cases throughout the manuscript, such as line 266 ' Fig.9 displays the effect of meteorological factors and emission reduction', please write clearly the numerical experiments, the time period and which domain for average etc.

Response: Thank you for your advice. The old Fig.9 is the current Fig.10. We have rewritten the sentence as "Fig.10 displays the effect of meteorological factors and emission reduction in Nanjing on air quality improvement during YOG (12-28 Aug., 2014). " (Line 351-352). And the caption of Fig. 10 (Line 359-261) has been changed as "Fig. 10. The simulated effect of meteorology and reduction on pollutant concentrations in Nanjing during the YOG (16-28 Aug. , 2014), Met. (Exp.2-Exp.3) represents the simulated effect of meteorology, while Red. (Exp.1-Exp.2) represents the simulated effect of reduction.". Besides, the details of numerical experiments were stated in section 2.3 Line 200-206.

Comment 9: The English in this manuscript should be carefully checked and much improved by correcting grammatical errors and rewording sentences, some of them are misleading and ambiguous.

Response: Thank you for your advice. The co-authors have helped to modify and improve the English in the manuscript carefully.

---

## Author Response (AR1)

Dear Editors and Reviewers,

Thank you very much for your letter and for the reviewers' comments concerning our manuscript entitled "Impacts of emission reduction and meteorological conditions on air quality improvement during the 2014 Youth Olympic Games in Nanjing, China" (doi:10.5194/acp-2017-114). Your comments are all valuable and very helpful for revising and improving our paper, as well as the important guiding significance to our researches. We have studied the comments carefully and have made corrections which we hope meet with approval. Based on the instructions, we have uploaded the file of the revised manuscript.

Appended to this letter is our point-by-point response to the reviewers' comments, the change list and the marked-up manuscript.

We would like to thank you for allowing us to resubmit a revised copy of the manuscript. We hope that the revised manuscript is accepted for publication in ACP.

Sincerely,

Qian Huang

**Responses to the reviewers' comments:**

Dear Reviewers,

Thank you very much for reviewing the manuscript and providing us the constructive comments and suggestions on our study. We have learned a lot from your advice and revised the manuscript. According to the Comment 5 from Reviewer 2, there might be some problem in Fig.8 in Section 3.3 (old manuscipt). And we carefully checked our data processing, simulation scenarios, and emission inventory, and found that the emission inventory used in Exp.1 (under emission control) had some problem with some points larger than those in the emission inventory before emission control. We're so sorry about that. We have corrected the problem, redone the model simulation of Exp.1, and reprocessed the data. And the model simulation data related to emission reduction are modified in the revised manuscript. Thank you very much for your understanding.

We have studied your comments carefully and have made corrections which we hope meet with approval. And point-by-point response to your comments are listed as below.

Sincerely,

Qian Huang

**Reviewer 1**

**Comment 1:** This manuscript described a study for the emission control scenario during the 2nd Youth Olympic Games in Nanjing using surface measurements and WRF/CMAQ model. This manuscript's English need improvement. It listed both model and measurement results, but it is not easy to track which is about observation and which is about the model. I suggest add something to make it clear. For instance, the title of section 3.1 can be "Observed air quality during YOG", and the section 3.2 changes to be "simulated impact of meteorological conditions".

**Response:** Thank you for your advice. We have followed your advice and modified the manuscript to make observation part and modeling part more clearly. The title of section 3.1 can be "Observed air quality during YOG", and the section 3.2 changes to be "simulated impact of meteorological conditions".

**Comment 2:** Another issue is that the discussions for the measurements and model are totally separated, and the modeled impact of $NO_x$ emission reduction on $O_3$ et al is not supported by the observation. Obviously the model or emission inventory has some biases, which should be addressed.

**Response:** Thank you for your advice. The measurements and model are separated because we want to investigate the variations of air pollutants from different point of view, one is the real reduction of air pollution from observation, the other is model reduction from meteorology and emission.

The observational decrease of $O_3$ may due to the meteorological conditions. In Aug. 2014, there were more overcast days, and the reduction in solar radiation could restrain the production of $O_3$. However, in the model simulation, underestimation of cloud cover could result in more solar radiation, which was conductive to the promotion of $O_3$. Besides, reduction of $NO_x$ could result in less titration of $O_3$ by $NO_x$, which also lead to higher simulation $O_3$. Thus, the observational $O_3$ variation and the model simulation $O_3$ variation are different, which was discussed in section 3.3.

**Comment 3:** Another issue is that this study did not discuss anything about emission and pollutant concentrations in surrounding areas, which sometimes can affect your results.

**Response:** Thank you for your comment. During the YOG holding month (Aug. 2014), though the surrounding area of Nanjing had taken part in the pollution emission control, their magnitudes of emission reduction were very small. And the emission reduction were mainly concentrated in the holding city, Nanjing, so we focus on the local emission and pollutant concentrations.

**Comment 4:** Page 1, line 27, "However, simulation" better to be "However, the model simulation"

**Response:** We are grateful for your suggestion. We have followed your advice and modified the manuscript (Line 27, Page1).

**Comment 5:** Page 1, line 28, "and raised $SO_2$" better to be "and could increase"

**Response:** We have modified it according to the comment (Line 29, Page1).

**Comment 6:** Page 2, line 48-49, "Preparatory work were carried out since 1 Jul., 2014" better to be "The preparation started from Jul. 1, 2014".

**Response:** We have changed it according to the suggestion (Line 48-49, Page 2).

**Comment 7:** Page 2, line 54-59. Please consider to split that long sentence to several sentences as it has grammar errors.

**Response:** Thank you for your advice. We have followed your suggestions and modify the manuscript. And the sentences (Line 52-57, Page 2) have been rephrased as follows:

Some local petrochemical, chemical and steel industries were forced to limit or halt their production. Coal-combustion enterprises were required to use high-quality coals with low sulfur content and ash content. And heavy pollution vehicles called "yellow label buses" were prohibited in Nanjing during 10-28 Aug.. Oil loading and unloading operations were strictly controlled. Construction process was forced to stop.

**Comment 8:** Page 4, line 137. "Exp.3 had the same inventory as Exp.2 but the weather" better to be "Exp.3 had the same inventory as Exp.2 but used the weather"

**Response:** We have changed it according to the suggestion (Line 203, Page 8).

**Comment 9:** Page 4, line 141. "meteorology on contaminants" better to be "meteorology on air quality".

**Response:** We have changed it according to the suggestion (Line 206, Page 8).

**Comment 10:** This manuscript should show a map of the emission reduction for Exp1 - Exp2, instead of just modeled concentration changes.

**Response:** Thank you for your advice. We have add some introduction of the emission inventory used in model simulation in Section 2.3 Emissions and simulation scenarios and offer the map (See Fig.2) of the emission reduction for Exp.1 - Exp.2.

**Comment 11:** Page 8, line 184, "most species had a good reflection", What does it mean?

**Response:** Thank you for your comment. We're sorry about the confusing sentence and have rewritten it. It means the concentrations of most species decreased obviously in Aug. 2014 compared with those in Jul. 2014 and Sept. 2014 (Line 249, Page 11).

**Comment 12:** Page 8, line 186-194. Please re-write to make it easy to understand.

**Response:** Thank you for your suggestion. We have rewritten the sentences (Line 251-260, Page 11).

**Comment 13:** Figure 6, 7 and the corresponding discussion in section 3.2. Are those comparisons for monthly averaged value, such as 10m wind, PBL heights? If so, please state it.

**Response:** Thank you for your comment. We have completed the captions of the figure (Line 325-328). And Figure 7 shows hourly average values of impact percentage (dspecies (%)= (Exp.2 - Exp.3)/Exp.2 * 100%) of $SO_2$, $NO_2$, $PM_{10}$, $PM_{2.5}$, CO, and $O_3$, respectively. To better display the bias of meteorological parameters, Figure 6 was replaced by Figure 8 in the revised manuscript (Line 329-334, Page 16), they're hourly average values.

[Figure]

**Fig. 8.** Bias of simulated hourly mean meteorological conditions during the YOG. (Bias = Meteorological Factors in 16-28 Aug., 2013 - Meteorological Factors in 16-28 Aug., 2014. (a) Bias of Wind at 10m during 16-28 Aug. (unit: m/s), (b) Bias of temperature at 2m during 16-28 Aug. (unit: K), (c) Bias of planetary boundary layer height during 16-28 Aug. (unit: m)).

**Comment 14:** Page 13, line 257-259. The $O_3$ increase should be due to the NOx emission reduction -> less titration.

**Response:** Thank you for your comment. We have corrected it (Line 340-342, Page 16) as follows:

"As for $O_3$, the variation was positive, especially in the downwind area of $NO_x$ heavy reduction region, which might due to the less titration of $O_3$ by $NO_x$"

**Comment 15:** Page 14, table 6. Why the modeled impact of the emission reduction on $NO_2$, $O_3$, $PM_{10}$, $PM_{2.5}$ diff significantly from the observations? You may discuss it.

**Response:** Thank you for your suggestion.

As discussed in Section 2.3 (Line 197-206), the simulation scenarios are reasonable. And the dynamic parameterization in WRF as well as the physical and chemical schemes of CMAQ applied in this research were the same as those in the research of Shu et al.(Shu is a member of our research group). The model performance has been validated by Shu et al., and they proved that WRF/CMAQ is reliable as shown in the following two pictures.

**Table 3.** Comparisons between the simulations and the observations at Shanghai, Nanjing and Hangzhou stations during 4–15 August 2013.

| Site[a] | Vars[b] | Mean | | $R$[e] | NMB[f] | RMSE[g] |
|---|---|---|---|---|---|---|
| | | OBS[c] | SIM[d] | | | |
| SH | $T_2$ (°C) | 33.27 | 31.38 | 0.91 | −5.68 % | 4.15 |
| | $RH_2$ (%) | 57.91 | 65.23 | 0.85 | 12.64 % | 19.3 |
| | $Wspd_{10}$ (m s$^{-1}$) | 4.59 | 4.66 | 0.77 | 1.53 % | 2.18 |
| | $Wdir_{10}$ (°) | 176.34 | 182.57 | 0.63 | 3.53 % | 41.44 |
| | $O_3$ (ppb) | 87.77 | 82.5 | 0.81 | −6.00 % | 38.79 |
| | $NO_2$ (ppb) | 29.01 | 38.25 | 0.54 | 31.85 % | 28.95 |
| NJ | $T_2$ (°C) | 32.95 | 30.98 | 0.84 | −5.98 % | 2.91 |
| | $RH_2$ (%) | 63.28 | 66.14 | 0.83 | 4.52 % | 9.41 |
| | $Wspd_{10}$ (m s$^{-1}$) | 3.21 | 3.4 | 0.74 | 5.92 % | 2.41 |
| | $Wdir_{10}$ (°) | 197.68 | 194.58 | 0.57 | −1.57 % | 71.19 |
| | $O_3$ (ppb) | 69.7 | 78.15 | 0.81 | 12.12 % | 36.8 |
| | $NO_2$ (ppb) | 41.44 | 40.09 | 0.61 | −3.26 % | 22.4 |
| HZ | $T_2$ (°C) | 33.25 | 31.08 | 0.8 | −6.53 % | 3.09 |
| | $RH_2$ (%) | 52.76 | 61.39 | 0.78 | 16.36 % | 13.96 |
| | $Wspd_{10}$ (m s$^{-1}$) | 3.04 | 3.32 | 0.75 | 9.21 % | 2.39 |
| | $Wdir_{10}$ (°) | 186.45 | 186.2 | 0.58 | −0.13 % | 69.44 |
| | $O_3$ (ppb) | 76.57 | 84.51 | 0.83 | 10.37 % | 33.95 |
| | $NO_2$ (ppb) | 31.06 | 27.21 | 0.66 | −12.40 % | 16.86 |

[a] Site indicates the city where the observation sites locate, including Shanghai (SH), Nanjing (NJ) and Hangzhou (HZ). [b] Vars indicates the variables under validation, including 2 m air temperature ($T_2$), 2 m relative humidity ($RH_2$), 10 m wind speed ($Wspd_{10}$), 10 m wind direction ($Wdir_{10}$), ozone ($O_3$) and nitrogen dioxide ($NO_2$). The words between the parentheses behind variables indicate the unit. [c] OBS indicates the observation data. [d] SIM indicates the simulation results from WRF/CMAQ. [e] $R$ indicates the correlation coefficients, with statistically significant at 95 % confident level. [f] NMB indicates the normalized mean bias. [g] RMSE indicates the root-mean-square error.

[Figure]

**Figure 6.** Hourly variations of the observed and the simulated $O_3$ concentrations in Shanghai, Nanjing and Hangzhou during 4 to 15 August 2013. The red solid lines show the modeling results, the black dot lines give the observations, and the solid gray lines represent the national standard for the hourly $O_3$ concentration, which is 200 µg m$^{-3}$.

Several factors contribute to the bias of $NO_2$, $O_3$, $PM_{10}$, $PM_{2.5}$ between simulation and observation. Firstly, a overall underestimation of emission reduction might result in less variations of $NO_2$, $PM_{10}$, and $PM_{2.5}$ caused by emission abatement. Secondly, the observational $O_3$ decreased during the emission control month (Aug. 2014) while the simulation $O_3$ increased slightly under emission control, which can be affected by the cloud simulation and the modeling chemical mechanism. During Aug. 2014, there were more overcast days, which may cause less solar radiation and was adverse for the promotion of $O_3$. However, the underestimation of modeling cloud cover could lead to higher simulation $O_3$. Besides, reduction of $NO_x$ could result in less titration of $O_3$ by $NO_x$, and overestimation of this chemical mechanism could also lead to higher simulation $O_3$.

What's more, the aim of Table 6 (Page 18) is to compare the simulated effect of meteorological conditions and emission reduction other than comparing the simulation with the observation. It want to indicate that the adverse meteorological conditions in Aug. 2014 could cause the increase of pollution concentrations while emission reduction could help to cut down the pollutants' ($SO_2$, $NO_2$, CO, $PM_{10}$, and $PM_{2.5}$) level during Aug. 2014.

**Reviewer 2**

**Comment 1:** line 112-113, 'the 9 state controlling air sampling sites in

Nanjing were chosen to represent the whole Nanjing'. Looking at fig 1b, I found 9 stations almost concentrate in the urban area, which is small compared with the whole Nanjing, so I doubt the 9 sites can represent the whole Nanjing, and it's better to collect some observations at rural sites of Nanjing for model validation.

**Response:** Thank you for your comment. Your suggestion is reasonable, but the limitation is that there are only 9 state controlling air sampling sites in Nanjing as shown in the paper. Among them, XL site is regarded as a suburban site, while CCM site is regarded as an urban site. Besides, the Nanjing Municipal Environmental Protection Bureau takes the 9 state controlling air sampling sites to represent the whole Nanjing and issues Nanjing Air Quality Daily Report. So we think it may be better to follow the local EPA. The details about choosing the sites have been added in Section 2.2 (Line 151-155, Page 5).

**Comment 2:** Line 149-160 presents the comparison between Aug. 2014 and Aug. 2013 and states' emission reductions did help the alleviation of air pollution......', you didn't look at and discuss the difference in meteorological conditions between the two years, how can you rule out the potential influence of meteorology, so please add meteorology comparison here. Also, there are evident emission reductions during Aug. 2014, with 22.1% for $SO_2$, 12.5% for $NO_x$, and 21.4% for $PM_{2.5}$, why the decrease in $PM_{2.5}$ concentration at CCM is just 9.8%, how about the proportion and relative changes of primary and secondary PM2.5?

**Response:** Thank you for your suggestion. We have reorganized this part (Line 222-228). And we have followed your advice and comparison of meteorological conditions is in Section 3.2.

The emission reduction percentages are the mean of the whole city, the details of emission reduction are added in Section 2.3 Line 173-195. However, the distributions of emission reduction are not uniform since the intensities of emission reduction are different from various trades. Also, the relationship between pollutant source and pollutant concentration is not linear. Thus, the decrease in $PM_{2.5}$ concentration at urban site CCM is not very big. The $PM_{2.5}$ observational data is total $PM_{2.5}$, so we can't distinguish the proportion and relative changes of primary and secondary $PM_{2.5}$.

**Comment 3:** Line 182-191, when comparing simulations in Aug. 2014 with that in Jul. and Sept. 2014, you try to say the pollutant concentrations declined with emission control, but rebounded after releasing control, however, the simulated $SO_2$ concentration in Aug. is larger than that in Jul. (5.1%), whereas $NO_2$ (19.8%) and CO (21.1%) in Aug. are larger than in Sept., how do you explain the larger $SO_2$, $NO_2$ an CO concentrations in Aug. although strict emission abatement is implemented than those in Jul. and Sept. with no emission reduction?

**Response:** Thank you for your comment. Firstly, this paragraph compares the observational data other than simulations in Aug. 2014 with that in Jul. and Sept. 2014. Secondly, we're sorry that there is a mistake in Line 190 (old manuscript): "the change percentage of species ($SO_2$, $NO_2$, $PM_{10}$, $PM_{2.5}$, CO and $O_3$) was -37.4%, 19.8%, -37.6%, -22.3%, 21.1%, and -47.2%, respectively at CCM station (Table 4)", "19.8%" should be "-19.8%", and we have corrected it (Line 259-260). Thirdly, Table 4 and Table 5 show the observational pollutants variations other than simulated pollutants variations in Jul., Aug. and Sept. at CCM station and XL station. We can see that at XL station (a suburban station), the concentrations of all species ($SO_2$, $NO_2$, $PM_{10}$, $PM_{2.5}$, CO, and $O_3$) in Aug. are the lowest compared to those in Jul. and Sept.. Besides, at CCM station (a urban station), the concentrations of most species ($NO_2$, $PM_{10}$, $PM_{2.5}$, and $O_3$) are the lowest compared to those in Jul. and Sept.. These show a pollutant concentration decline trend after emission control and a rebound trend after releasing control. Besides, at CCM station, the observational $SO_2$ concentration in Aug. is larger than that in Jul. (5.1%), whereas CO (21.1%) in Aug. are larger than in Sept., which could be caused by many factors, such as traffic and other unpredictable emissions around the site. As for traffic control, only the heavy pollution vehicles called "yellow label buses" were prohibited in Nanjing during 10-28

Aug.. To meet the traffic demand of numerous tourists, athletes, and freightage, there could be more traffic pollution and raised the level of $SO_2$, $NO_x$ and CO. Besides, $NO_x$ was mainly emitted from power plants, so the overall NOx was the lowest during the emission control month. However, they don't bother the overall variation trend of the six species.

**Comment 4:** line 227-228, 'Consequently, Exp.2 resulted in higher pollutant concentrations for all species as shown in Fig.7', this is not true, although the domain averages of pollutant values increase from Exp3 to Exp2, it is apparent that the spatial distribution did not show a consistent increase in the domain, such as the large decreases in all components but $O_3$ to the northeast, and the decreases in $SO_2$, $NO_2$, CO, $O_3$ in portions of Nanjing, so the meteorological condition in Aug. 2014 did not necessarily lead to increases in pollutant levels, so I suggest more discussion on the different responses to meteorology in the domain with analysis of meteorological variable changes.

**Response:** Thank you for your comment. This paper tries to discuss the overall impact of meteorological conditions. Based on Fig.7, statistics show that meteorology in Aug. 2014 led to total increases in pollutant levels. Line 301-302 offer the details : "For $SO_2$, $NO_2$, $PM_{10}$, $PM_{2.5}$, CO, and $O_3$, their concentrations were increased by 17.5%, 16.9%, 19.0%, 19.5%, 7.8% and 0.8%". Factors such as topography could affect locally, and may cause discontinuous increases in Fig.7 , but it did not affect the overall increase trend. So, partial decrease is not that important. The analysis of meteorological variable changes was in Line 303-314.

**Comment 5:** section 3.3, similar problems in this section, when emission reduction lead to apparent decreases in concentrations of all pollutants except $O_3$, how do you explain the apparent increases in the southern parts of Nanjing (Fig. 8)? are there some feedbacks among aerosols, radiation (photolysis), cloud and consequent effects on chemical processes, please elaborate on mechanisms behind these changes instead of just presenting model results.

**Response:** Thank you for your comment. The mechanisms about meteorology have been discussed in Section 3.2. In Section 3.3, the apparent increases of ($SO_2$, $NO_2$, $PM_{10}$, $PM_{2.5}$ and CO) in the southern parts of Nanjing seems unreasonable. To find out the reason, we carefully checked our data processing, simulation scenarios, and emission inventory. We found that the problem was not caused by meteorology but the emission inventory. The emission inventory used in Exp.1 (under emission control) had some problem with some points larger than those in the emission inventory before emission control. The emission under control should not exceed the emission before control. We are sorry about that. We have corrected the emission inventory (under emission control), redone the model simulation of Exp.1 and reprocessed the data. And the corrected figure (See Section 3.3, Fig.9) as shown below don't have increases (SO₂, NO₂, PM₁₀, PM₂.₅ and CO) in the southern of Nanjing. Besides, emission reduction led to completely decrease (SO₂, NO₂, PM₁₀, PM₂.₅ and CO) in the whole city, and increase of O₃ in Nanjing. The drop of O₃ was due to the reducing NO₂ and less titration impacts.

[Figure]

**Fig. 9.** Influence of emission reduction on hourly mean concentrations of pollutants in Aug. 2014. (Black thick lines draw the outline of Nanjing. Picture a - f are hourly average values of impact percentage (dspecies (%) = (Exp.1 - Exp.2)/ Exp.2*100%) of SO₂, NO₂, PM₁₀, PM₂.₅, CO and O₃, respectively.).

**Comment 6:** Regarding Fig.9, please explain how the meteorological change lead to day-to-day variations (either increase or decrease) of pollutant concentration.

**Response:** Thank you for your comment. The old Fig.9 is the current Fig.10 in section 3.4, it aims to compare the simulated effect of meteorology and emission reduction from day to day during the YOG other than to explain how meteorological change lead to day-to-day variations. From Fig.10 , we can see that emission control caused decreases of pollutant ($SO_2$, $NO_2$, $PM_{10}$, $PM_{2.5}$, and CO) concentration while meteorology caused increases of pollutant ($SO_2$, $NO_2$, $PM_{10}$, $PM_{2.5}$, CO and $O_3$) concentration in most of the time.

**Comment 7:** Some tables like Table 4 and 5 can be removed because this manuscript is not a data report.

**Response:** Thank you for your advice. Table 4 and Table 5 are statistical analysis of observational data, we think they're important and retain them may be better.

**Comment 8:** Please describe clearly the spatial and time scales of the presented data or model results and the comparison between cases throughout the manuscript, such as line 266 ' Fig.9 displays the effect of meteorological factors and emission reduction' , please write clearly the numerical experiments, the time period and which domain for average etc.

**Response:** Thank you for your advice. The old Fig.9 is the current Fig.10. We have rewritten the sentence as "Fig.10 displays the effect of meteorological factors and emission reduction in Nanjing on air quality improvement during YOG (12-28 Aug., 2014). " (Line 351-352). And the caption of Fig. 10 (Line 359-261) has been changed as "Fig. 10. The simulated effect of meteorology and reduction on pollutant concentrations in Nanjing during the YOG (16-28 Aug. , 2014), Met. (Exp.2-Exp.3) represents the simulated effect of meteorology, while Red. (Exp.1-Exp.2) represents the simulated effect of reduction.". Besides, the details of numerical experiments were stated in section 2.3 Line 200-206.

**Comment 9:** The English in this manuscript should be carefully checked and much improved by correcting grammatical errors and rewording sentences, some of them are misleading and ambiguous.

**Response:** Thank you for your advice. The co-authors have helped to modify and improve the English in the manuscript carefully.

**Reviewer 3**

**General Comments:**

**Comment 1:** This paper tried to evaluate the impacts of emission reduction and meteorologicalconditions on the air quality improvement during an air pollution control period-YOG of Nanjing. Accurate quantification of the influence of emission reduction and meteorological conditions is important to evaluate the air pollution control measures. This paper used both observation data and modeling results to address this issue. However, this manuscript has major writing and structure problem. The validation of model simulation and uncertainty analysis is essential and required but lack in the manuscript.

**Response:** Thank you for your comment. We have extended the model description part in Section 2.2, in this part, we explained that the dynamic parameterization in WRF as well as the physical and chemical schemes of CMAQ applied in this research were the same as those in the research of Shu et al. (Shu is a member of our research group) and were proven to have good simulation performance. So we no longer validate the model performance and uncertainty in this paper.

The following table and figure are the evaluation of WRF/CMAQ performance from Shu et al. (2016). The table presents the performance statistics, including the values of R, the NMB and the RMSE, which are all calculated for 2 m air temperature ($T_2$), 2 m relative humidity ($RH_2$), 10 m wind speed ($Wspd_{10}$), 10 m wind direction ($Wdir_{10}$), surface $O_3$ concentrations and surface $NO_2$ concentrations in Shanghai (SH), Nanjing (NJ) and Hangzhou (HZ), China. As indicated in the table, the simulated results of surface air temperature and relative humidity from WRF show good agreement with the observations. The highest correlation coefficient of $T_2$ is found to be 0.91 at SH, followed by 0.84 at NJ and 0.80 at HZ (statistically significant at 95 % confident level). The corresponding correlation coefficients for $RH_2$ are 0.85, 0.83 and 0.78, respectively. The values of RMSE for $T_2$ at SH, NJ and HZ are 4.15, 2.91

and 3.09 ◦C and those for $RH_2$ are 19.3, 9.41 and 13.96 %, respectively. The simulation underestimates $T_2$ and overestimates $RH_2$ to some certain extent, however, they're reasonable and acceptable compared to some relevant studies. Besides, the table indicates that the simulation of $Wspd_{10}$, $Wdir_{10}$, and concentrations of pollutants are also reliable. The following figure shows the comparisons between the modeling results from CMAQ and the observed hourly concentrations of $O_3$ in Shanghai, Nanjing and Hangzhou during 4-15 Aug. 2013. Obviously, the observations and the simulated results present reasonable agreement at each site, with the correlation coefficients of 0.81 to 0.83, NMB of −6 to 12.12 % and RMSE of 33.95 to 38.79 ppb. Moreover, the simulation also reproduces the diurnal variation of $O_3$, which shows that the concentration reaches its maximum at around noontime and gradually decreases to its minimum after midnight.

**Table 3.** Comparisons between the simulations and the observations at Shanghai, Nanjing and Hangzhou stations during 4–15 August 2013.

| Site[a] | Vars[b] | Mean | | R[e] | NMB[f] | RMSE[g] |
|---|---|---|---|---|---|---|
| | | OBS[c] | SIM[d] | | | |
| SH | $T_2$ (°C) | 33.27 | 31.38 | 0.91 | −5.68 % | 4.15 |
| | $RH_2$ (%) | 57.91 | 65.23 | 0.85 | 12.64 % | 19.3 |
| | $Wspd_{10}$ (m s$^{-1}$) | 4.59 | 4.66 | 0.77 | 1.53 % | 2.18 |
| | $Wdir_{10}$ (°) | 176.34 | 182.57 | 0.63 | 3.53 % | 41.44 |
| | $O_3$ (ppb) | 87.77 | 82.5 | 0.81 | −6.00 % | 38.79 |
| | $NO_2$ (ppb) | 29.01 | 38.25 | 0.54 | 31.85 % | 28.95 |
| NJ | $T_2$ (°C) | 32.95 | 30.98 | 0.84 | −5.98 % | 2.91 |
| | $RH_2$ (%) | 63.28 | 66.14 | 0.83 | 4.52 % | 9.41 |
| | $Wspd_{10}$ (m s$^{-1}$) | 3.21 | 3.4 | 0.74 | 5.92 % | 2.41 |
| | $Wdir_{10}$ (°) | 197.68 | 194.58 | 0.57 | −1.57 % | 71.19 |
| | $O_3$ (ppb) | 69.7 | 78.15 | 0.81 | 12.12 % | 36.8 |
| | $NO_2$ (ppb) | 41.44 | 40.09 | 0.61 | −3.26 % | 22.4 |
| HZ | $T_2$ (°C) | 33.25 | 31.08 | 0.8 | −6.53 % | 3.09 |
| | $RH_2$ (%) | 52.76 | 61.39 | 0.78 | 16.36 % | 13.96 |
| | $Wspd_{10}$ (m s$^{-1}$) | 3.04 | 3.32 | 0.75 | 9.21 % | 2.39 |
| | $Wdir_{10}$ (°) | 186.45 | 186.2 | 0.58 | −0.13 % | 69.44 |
| | $O_3$ (ppb) | 76.57 | 84.51 | 0.83 | 10.37 % | 33.95 |
| | $NO_2$ (ppb) | 31.06 | 27.21 | 0.66 | −12.40 % | 16.86 |

[a] Site indicates the city where the observation sites locate, including Shanghai (SH), Nanjing (NJ) and Hangzhou (HZ). [b] Vars indicates the variables under validation, including 2 m air temperature ($T_2$), 2 m relative humidity ($RH_2$), 10 m wind speed ($Wspd_{10}$), 10 m wind direction ($Wdir_{10}$), ozone ($O_3$) and nitrogen dioxide ($NO_2$). The words between the parentheses behind variables indicate the unit. [c] OBS indicates the observation data. [d] SIM indicates the simulation results from WRF/CMAQ. [e] R indicates the correlation coefficients, with statistically significant at 95 % confident level. [f] NMB indicates the normalized mean bias. [g] RMSE indicates the root-mean-square error.

[Figure]

**Figure 6.** Hourly variations of the observed and the simulated $O_3$ concentrations in Shanghai, Nanjing and Hangzhou during 4 to 15 August 2013. The red solid lines show the modeling results, the black dot lines give the observations, and the solid gray lines represent the national standard for the hourly $O_3$ concentration, which is $200 \, \mu g \, m^{-3}$.

**Comment 2:** The paper lacks in-depth discussions of the observation data and model results. Some conclusions are too arbitrary and lack sufficient evidence to back the interpretations of the results (see detail comments below).

**Response:** Thank you for your comment. This paper tries to apply model simulations to investigate the reason why observation pollutant concentration changes. We have studied your comment and added some discussion in-depth about the results in the revised manuscript.

**Comment 3:** The literature review in the introduction section needs improvement.

**Response:** Thank you for your comment. According to your detail comments, we have improved the introduction section in the revised manuscript.

**Comment 4:** The quality of English needs substantial improving. I believe that the paper needs substantial revisions before considering to be published at ACP.

**Response:** Thank you for your comment. The co-authors have helped to improve the English of the paper, and some sentences have been rewritten and reorganized.

**Detail Comments**

**Comment 1:** Line 22-26 This sentence here is not rigorous. What concentration? Hourly average? Daily average? From what data? Observation data at which site? You'd better give the standard deviations of the data.

**Response:** Thank you for your comment. We're sorry about the ambiguous expression. They're the hourly average observational concentrations. And they are the mean of the two representative sites in Nanjing. We have rewritten the sentences as follows:

"During the YOG holding month (Aug., 2014), the hourly mean observational concentration of $SO_2$, $NO_2$, $PM_{10}$, $PM_{2.5}$, CO and $O_3$ was 11.6 $\mu g/m^3$, 34.0 $\mu g/m^3$, 57.8 $\mu g/m^3$, 39.4 $\mu g/m^3$, 0.9 $mg/m^3$, and 38.8 $\mu g/m^3$, respectively, which were below China National Ambient Air Quality Standard."

Besides, we have added some explanation in Section 3.1 Observed air quality during YOG (See Line 214-217, Page9). And the standard deviations of the data was given in Section 3.1.

**Comment 2:** The introduction section should be rewritten and reorganized. The references cited in the introduction section should be more targeted and well selected. Take Line 78-82 for example, the references cited here have nothing to do with the topic of the paper. Line 60-82, too many references are cited without summary and in-depth understanding.

**Response:** Thank you for your comment. The references cited in the introduction section are mainly discussing the impact of emission reduction or meteorology on air quality in social events, like Beijing Olympic Games, the 16th Asian Games in Guangzhou and the World Expo in Shanghai. And some discussed the air pollution characteristics in Yangtze River (where Nanjing locates). All of them have reference value to our research. Line 78-79 "Xu et al. (2013) concluded that $PM_{2.5}$ was mainly emitted from anthropogenic sources other than biogenic sources." related to the impact of emission reduction, it indicated that cut down anthropogenic sources could decrease PM2.5 in the air. Line 79-80 "Dong et al. (2013) found that independent $NO_x$ emission reduction would strengthen $O_3$ as a side effect in YRD." helped to explained the simulated increase of $O_3$ in our research. Some introduction of references might be simple or not very important, we have modified them and added some references. The modified sentences are in Line 60-90.

**Comment 3:** Section 2.2, Fig.1 is hard to read. The authors stated that the 9 stations were chosen for representing the whole Nanjing city. But all of the 9 stations located at the center part of the city. I doubt can they represent the whole city? Moreover, what is the purpose of these sites? For model validation? Please give the results of model validation.

**Response:** Thank you for your comment.

There are only 9 state controlling air sampling sites in Nanjing as shown in this paper. They locate in different districts of Nanjing. And the density of population, traffic conditions and economics can differ a lot in different district, for example, the urban district Gulou District (where CCM station locates) and suburb district Xianlin District (where XL station loctes).

In this condition, Nanjing Municipal Environmental Protection Bureau chooses the local 9 state controlling air sampling sites to represent the whole Nanjing city. In conformity with this, we chose the 9 state controlling air sampling sites to represent the whole Nanjing while analyzing model simulation impacts.

Thus, they're not use for model validation. The details about model validation have been answered in the General Comment Response part (General Comments, Comment 1).

Besides, we have added the reason why we choose the 9 sites in the revised manuscript (Line 151-155, Page6).

**Comment 4:** Section 2.3 The description here is quite ambiguous. Which year of the emission inventory is used for simulation? How do the authors make the emission inventory after reduction? How to determine the reduction ratio? Based on the control measures? Is there any hypothesis here? If there is hypothesis? What is the uncertainty? Please state the experimental process in detail.

**Response:** Thank you for your advice. We have added the detail about how the innermost domain emission inventories were set in Section 2.3 (Line 173-187) in the revised manuscript.

The inventory before emission control was based on the local emission in 2012. According to the control measures offered by the local Environmental Protection Agency (EPA), we made the emission inventory under emission control. The emission control measures include all coal-combustion enterprises must use high-quality coals with low sulfur content less than 0.5% and ash content less than 13%, over 100 local petrochemical, chemical and steel enterprises were forced to cut or halt their production during Aug. 2014, and heavy pollution vehicles called "yellow label buses" were prohibited. And more details about emission control measures like the reduction ratios of some enterprises were in *2014 Youth Olympic Games Nanjing Environmental Air Quality Assurance and Emergency Response Program* offered by the local EPA. So, there could still be some bias with the emission inventories used in model simulation.

**Comment 5:** Section 3.1 The title of this section is inconsistent with the content. Why CCM and XL station are chosen for study? Can they represent the whole study area of the modeling or Nanjing (same as detail comments NO.3)? The data analysis in this section should be more rigorous and more in-depth. Line 147-148 How to get the reduction percentage? Calculate from observation data or other ways? Line 154-156 Why the authors avoid discussion of $NO_2$ at CCM and CO at XL? Line 157-158 The discussion here is inaccurate. The deviation of $PM_{10}$ and $PM_{2.5}$ is larger in 2014. Line 158-160 How to get this conclusion from the analysis above? Line 182-199 Similar problems as above. Line 190-191 The change percentage of $NO_2$ listed here is 19.8 %, but in Table 4 is -19.8%, please check the correctness and consistency of your results. In line 193-194, the authors said that "the pollutant concentrations declined with emission control, but rebounded after releasing control". How to explain the higher simulated concentrations of $SO_2$ and CO during Aug. with strictly control measures? The authors listed too many tables in this section without in-depth analysis and solid discussions.

Response: Thank you for your comment. To avoid misunderstanding, we have changed the title as 3.1 Observed air quality during YOG. And the reason why we choose CCM and XL station for study has been added in Section 2.1 Data description (Line 109-123, Page 4). Both of the two stations are state controlling air sampling sites. The data quality assurance and quality control procedures for monitoring strictly follow the national standards (State Environmental Protection Administration of China, 2006). Caochangmen (CCM) Station (118.75°E, 32.06°N) locates in Gulou District, the city center of Nanjing. Gulou District is the center of economy, politics, culture and education in Nanjing. Here gathers many East China's high-end industrial and corporate headquarters. Besides, over 90% provincial authorities, more than 20 colleges and universities, and more than 120 research institutes situate in Gulou District. It's the most populated area in Nanjing, with lively commercial hub and heavy traffic. Thus, CCM station was chosen to represent the urban status of Nanjing. The other station is Xianlin (XL) Station (118.92°E, 32.11°N ). XL station locates in Qixia District, the suburb of Nanjing. Compared to Gulou District, Qixia District is much more sparsely populated. And there is no traffic congestion problem in Qixia District. Thus, XL station was chosen to represent the suburban status of Nanjing. The reduction percentages were percentages of the emission sources, the details about the emission reduction were added in Section2.3 Emissions and simulation scenarios (Line 173-195). In the revised manuscript, to prevent misunderstanding, we no longer mention the emission reduction percentage in Section 3.1.

In order to stressed the observational concentration of most species decreased in Aug. 2014, we didn't mentioned the slightly rise of $NO_2$ at CCM and CO at XL. The slightly rise of $NO_2$ and CO could be caused by traffic. To meet the traffic demand of numerous tourists, athletes, and freightage, there could be more traffic pollution and raised the level of $NO_2$ and CO.

Thanks for your correction. Line 157-158 (old manuscript) is inaccurate, the conclusion is not reasonable in Line 158-160 (old manuscript). we have corrected them (See Line 223-225, Page 9) as "Besides, the smaller standard deviation (std) of $SO_2$, $NO_2$, CO and $O_3$ revealed that concentrations of these air pollutants varied more steadily in Aug. 2014. The drop of pollutant concentration could be caused mainly by meteorology conditions or emission reductions. And we will discuss the reason based on model simulations in Section 3.2 and Section 3.3.". Besides, we have corrected the problem in Line 182-199 (old manuscript), the details are in Line 262-269.

Sorry about the error in Line 190-191 (old manuscript), the change percentage of $NO_2$ should be -19.8% other than 19.8%, and we have corrected it.

The discussion of this part has been rewritten (Line 249-269). According to Table 4 and Table 5, concentrations of most species decreased in Aug. 2014, but rebound in Sept. 2014. Besides, the simulated concentration of $SO_2$ and CO during Aug. were not higher. The simulated $SO_2$ dropped by 24.6% and the simulated CO dropped by 7.2% (See Section 3.3 Simulated impact of emission reduction, Line 338-340 ) .

**Comment 6:** Line  221-232, The authors should  avoid ambiguous discussion.  The word such as "lower temperature and weaker winds", "rather worse meteorological conditions" is quite obscure to readers.

Line 227, The authors stated "······ which was consistent with the observations", could you give more detailed comparison results of model and observations? How about the accuracy of the simulated meteorological parameters? Fig. 6, What do "data1" and "data2" stand for?

**Response:** Thank you for your comment. We have rewritten this section (Section 3.2 Simulated impact of meteorological conditions Line 291-328). In the revised manuscript, the bias of meteorological parameters during the two simulated period were added to explain the different weather conditions. Details about the model performance please see the earlier response to General Comments, Comment 1. "data1" and "data2" means nothing, they should not show in the figure. We're sorry about the mistake, and have redrawn the figure.

**Comment 7:** How to explain the spatial distributions of the impact percentage? For CO and $O_3$, the simulated concentrations of Exp.2 are lower than those of Exp.3, especially for the north part of Nanjing city.

**Response:** Thank you for your comment. "For CO and $O_3$, the simulated concentrations of Exp.2 are lower than those of Exp.3" is incorrect. And $O_3$ of Exp.2 was not higher in the north part of Nanjing (Fig.7). Statistically, as for the mean of the whole city and the mean of 9 sites, the simulated CO and $O_3$ concentrations of Exp.2 are higher than those of

Exp.3. As discussed in Line 302, CO and $O_3$ were increased by 7.8% and 0.8% (the mean of 9 sites). The decrease of CO in the small section of northern Nanjing didn't bother our conclusion, we are more concern about the overall impact.

**Comment 8:** Section 3.3, Line 247-248, the statement here is ambiguous. 9.2% and 38.1% is from model results or others? 9.2% to 38.1% is a fuzzy range. Line 249-250, what do you mean? What is the definition of short-lived chemical composition? Line 250-251 How to explain the uneven distribution of the impact percentage? Line 256-257 The reduction ratios here are compared to what period? The authors should give more exact time during the discussion.

**Response:** Thank you for your suggestion. We're sorry about the ambiguous expression. 9.2% and 38.1% is from emission inventories used for simulations before and after emission control. In the revised manuscript, we have added description about emission inventories in Section 2.3, so we no longer refer the cutting ratios of emission inventories in Section3.3. Line 249-251 (old manuscript) means the uneven distribution of impact percentage was related to the uneven distribution of emission sources and the uneven reduction of emission sources. Thus the large simulation variations (Exp.1 - Exp.2) occurred in the west of Nanjing corresponding to the downwind regions of heavy reduction districts was reasonable.

Short-lived chemical composition refers to the chemical composition whose residence time in the atmosphere is short.

Line 256-257 (old manuscript) The reduction ratios referred to the simulated pollutant concentrations before and after emission control in Aug. 2014 ( the holding month of YOG). And the exact time was explained in Line 340.

**Comment 9:** Section 3.4 Why do you choose 16th Aug. to 28th Aug. not the whole month of Aug. as the study time here? Line 270-271 How can you make the conclusion here? From Fig. 9, it seems that the influence of meteorological conditions is more important for the air quality of Nanjing. Line 278-291 The authors focus on discussing difference of emission reduction influence at two sites. However, 0.9 %, 1.1 % etc. is quite small change. What is the result when considering the uncertainty of the model simulations? Line 299-308 The discussions here lack of evidence.

**Response:** Thank you for your comment. The old Fig.9 is the current Fig.10. The holding time of YOG is 16th Aug. to 28th Aug., to highlight the impact during the holding time, we choose 16th Aug. to 28th Aug..

As discussed in Section 3.2 Simulated impact of meteorological conditions, meteorological conditions in Aug. 2014 led to increases of pollutant levels compared to those under the conditions in Aug. 2013.

However, the discussion of observational data in Section 3.1 showed that the observational pollutant concentrations were lower in Aug. 2014 compared to those in Aug. 2013. So, we could conclude that the weather conditions in Aug. 2014 were worse than those in Aug. 2013 and might raise the pollutant level, the observational drop of pollutant concentrations in Aug. 2014 compared to those of Aug. 2013 was mainly due to emission reduction.

0.9% and 1.1% (old manuscript) are simulated impact percentage of $O_3$ at sites. Though they're small , they reflected that meteorology in Aug. 2014 could raise $O_3$, and emission reduction could also raise $O_3$ considered the reducing $NO_2$ and the titration effect. And they still support our conclusion. The details about the simulation model performance are in the previous response (General Comments, Comment 1).

Discussions in Line 299-308 (old manuscript) were according to the emission control measures as introduced in Section 2.3 Line 178-187.

**Technical Comments**

**Comment 1:** The authors should refer to "the guidelines for authors" of ACP to prepare the manuscript.

**Response:** Thank you for your comment. We have carefully read "the guidelines for authors" of ACP and revised the manuscript.

**Comment 2:** Abbreviations should be given for the first time. Such as "CST" etc.

**Response:** Thank you for your comment. Sorry about our carelessness. We have revised the manuscript (See Line 197, Page8 "The simulated period was from 27 Jul. to 1 Sept. (China standard time, CST)" ). And "CST" means China standard time in this paper.

**Comment 3:** The date format need to be uniform.

**Response:** Thank you for your suggestion. We have revised the date format to be uniform.

**Comment 4:** Spaces must be included between number and unit.

**Response:** Thank you for your suggestion. We have checked and revised the manuscript.

**Comment 5:** Fig. 9 The legend makers "Met" and "Red" here are easy to lead misunderstanding. You'd better use "Met." and "Red.".

**Response:** Thank you for your suggestion. We have corrected them.

**Comment 6:** The reference format should be uniform. Too many references in Chinese are cited.

**Response:** Thank you for your suggestion. We have checked and correct the format. Besides, we have added some more references in English and cut some references in Chinese.

**Comment 7:** The English of this manuscript needs substantial improvement.

**Response:** Thank you for your comment. The co-authors have helped to improve the English of the manuscript and some sentences have been rewritten and reorganized.

**Change list:**

**1. Line 21, 24-32:** Rephrase the Abstract section.

**2. Line 45-46:** Cite one more reference.

**3. Line 48-58, 60-63, 74-92, 99:** Rephrase the Introduction section.

**4. Line 109-123:** Rephrase Section 2.1 Data description.

**5. Line 127-143, 146-158:** Rephrase Section 2.2 Model description.

**6. Line 173-195:** Add some details about emission inventories used in model simulations.

**7. Line 197, 199, 203, 206:** Rephrase the sentences.

**8. Line 209-212, 214-217, 221-225, 248-249, 251-260, 262-265, 267-269:** Rephrase Section 3.1 Observed air quality during YOG.

**9. Line 291-294, 297-300, 303-328:** Rephrase Section 3.2 Simulated impact of meteorological conditions.

**10. Line 330:** Change the Section name as "3.3 Simulated impact of emission reduction".

**11. Line 335, 338-346:** Rephrase Section 3.3 Simulated impact of emission reduction.

**12. Line 350:** Change the Section name as "3.4 Comparison of simulated meteorological factors and emission reduction".

**13. Line 351-352, 360-361, 369-370, 373-376:** Rephrase Section 3.4 Comparison of simulated meteorological factors and emission reduction.

**14. Line 378-383:** Change Table 6 and its caption.

**15. Line 390, 392:** Rephrase Section 3.4.

**16. Line 401, 404-405:** Rephrase Section 4 Summary and conclusions.

**17. Delete some references as listed below:**

[revised manuscript text omitted]

---

## Referee Report (RR1)

The authors have largely improved the manuscript by addressing my comments and revising the manuscript accordingly, most of the questions have been clarified and additional model simulations were conduced and interpreted, with some new interesting results presented. I recommend acceptance of this manuscript after minor revisions on English and tables/figures.

---

## Author Response (AR2)

Dear Editors and Reviewers,

Thank you very much for your letter and for the reviewers' comments concerning our manuscript entitled "Impacts of emission reduction and meteorological conditions on air quality improvement during the 2014 Youth Olympic Games in Nanjing, China" (doi:10.5194/acp-2017-114). Your comments are all valuable and very helpful for revising and improving our paper, as well as the important guiding significance to our researches. We have investigated the comments carefully and made corrections which we hope to meet with approval. Based on the instructions, we have uploaded the file of the revised manuscript.

Appended to this letter is our point-by-point response to the reviewers' comments, the change list and the marked-up manuscript.

We would like to thank you for allowing us to resubmit a revised copy of the manuscript. We hope that the revised manuscript can be accepted for publication in ACP.

Sincerely,

Qian Huang

**Responses to the reviewers' comments:**

Dear Reviewers,

Thank you very much for reviewing the manuscript and providing us the constructive comments and suggestions on our study. We have learned a lot from your advice and revised the manuscript. There were some problems with the statistics of simulated particulate matters in the old manuscript, and we have corrected them in the revised manuscript. Thank you very much for your understanding.

We have studied your comments carefully and have made corrections which we hope meet with approval. And point-by-point response to your comments are listed as below. Besides, the appendixes are the change list and the marked-up manuscript.

Sincerely,

Qian Huang

**Reviewer 1**

**Comment 1:** This manuscript added some contents after the revision. However, its English still needs improvements.

**Response:** Thank you for your advice. The co-authors have helped to modify and improve the English in the manuscript carefully.

**Reviewer 2**

**Comment 1:** I don't think the 9 sites around downtown can represent the whole Nanjing, but I understand the possible limitation of available observation for this study.

**Response:** Thank you for your comment. Nanjing is a highly urbanized city. Up to now, there are only 9 state air monitoring sites in Nanjing in total. The 9 sites include urban sites and suburban sites. Besides, the Nanjing Environmental Protection Bureau takes the 9 sites to represent the whole Nanjing and releases online official air quality data every day. Thus, we have no choice but to choose the state air monitoring sites to represent the whole city.

**Comment 2:** Although only $PM_{2.5}$ observational data is presented in this study, I still suggest you present model simulated changes in both primary and secondary $PM_{2.5}$ due to emission reduction, which may help explain the weak $PM_{2.5}$ response (just 9.8%), I also think that discussion about $PM_{2.5}$ components will make the manuscript to be more comprehensive.

**Response:** Thank you very much for your advice. We have added the discussion about model simulated changes in both primary and secondary particulate matters (PM) including $PM_{10}$ and $PM_{2.5}$ in Section 3.2-3.4. And in the revised manuscript, Fig.11, Fig.12, and Table 6 show the effects of emission reduction on primary and secondary PM, and suggest that emission control has much greater impacts on primary PM than on secondary PM during Aug. 2014.

**Comment 3:** How to explain the increase of $SO_2$ concentration (5.1%) although $SO_2$ emission is reduced by 22.1% in August compared with July 2014? what's the meaning of "unpredictable emissions"? based on the difference between the two sensitivity runs, the increasing $SO_2$ concentration with reduction of $SO_2$ emission looks strange, please give more discussion here.

**Response:** Thank you for your comment. The cutting percentage of $SO_2$ emission was 25.0% for the whole city. However, the emission reduction was inhomogeneous in the city, which could be larger in non-urban areas and smaller in urban areas. This may be one of the reasons for explaining the higher $SO_2$ concentration at CCM station in Aug. compared to Jul. (5.1%). For the sensitivity runs, the $SO_2$ emissions reduction really leads to the decrease of $SO_2$ concentration.

**Comment 4:** " This paper tries to discuss the overall impact of meteorological conditions ...... partial decrease is not that important". I am not satisfied with the response, although $SO_2$, $NO_2$, $PM_{10}$, $PM_{2.5}$, CO, and $O_3$ concentrations increased in terms of domain average, there are large areas of concentration decrease for these species, especially for $O_3$, which require a detailed analysis of these changes in response to variations of meteorological variables and chemical reactions (such as temperature, cloud, PBL etc.)

**Response:** Thank you for your comment. We have added detailed discussions about the spatial pollutant changes in response to variations of meteorological variables and chemical reactions in revised Section 3.2. For $SO_2$, $NO_2$, $PM_{10}$, $PM_{2.5}$, CO, and $O_3$, their levels were increased in Aug. 2014 in terms of the city mean. Besides, for $SO_2$, $NO_2$, $PM_{10}$, $PM_{2.5}$, CO, $PM_{10p}$, $PM_{10s}$, $PM_{2.5p}$, and $PM_{2.5s}$, there were some small decreasing areas in the northeast Nanjing, which could be caused by the effect of predominant winds. In domain 4, the simulated predominant wind was northeast wind in Aug. 2014, while that was southeast wind in Aug. 2013. So, the simulated diffusion condition of northeast Nanjing might be better in Aug. 2014. For $O_3$, increasing concentrations were shown in northern and eastern Nanjing, while decreasing concentrations occurred in some southern areas during Aug., 2014, which corresponded well to the distribution of cloud cover. Cloud cover could affect the production of ozone by affecting radiation.

**Comment 5:** The authors corrected errors in Fig. 8 by reruning the model using corrected emission inventory.

**Response:** Thank you for your comment.

**Comment 6:** From the figure 10, a clear impression is emission reduction has little effect on reducing $PM_{2.5}$ ($PM_{10}$) level in Nanjing, which appear not to support the conclusion in this manuscript " emission reduction is the dominant factor of the air quality improvement during the YOG". Besides, in fig. 10, the changes due to emission reduction is hardly to see for species other than $SO_2$ and $NO_x$, is it possible to use different scales for the changes from meteorology and emission reduction?

**Response:** Thank you for your comment. We use "very important" instead of "dominant" in the revised manuscript. We have revised Fig. 12 (the original Fig. 10) in Section 3.4 and added more species ($PM_{10p}$, $PM_{10s}$, $PM_{2.5p}$, and $PM_{2.5s}$) , which show day-to-day simulated effect of meteorology and emission reduction effects during the whole month. As you can see, most of the time, meteorological conditions and emission reduction had opposite effects on pollutants ($SO_2$, $NO_2$, $PM_{10}$, $PM_{2.5}$, CO, $PM_{10p}$, $PM_{10s}$, $PM_{2.5p}$, and $PM_{2.5s}$), especially during the YOG (16-28, Aug., 2014). Emission control always played a positive role and cut down the pollutant levels during the whole month, while weather conditions could play a negative role. Table 6 (in Section 3.4) illustrated the opposite effects. These all indicated that emission reduction is the very important factor for air quality improvement during the YOG. Though, the emission reduction percentage was 42.8% for $PM_{10}$, 36.2% for $PM_{2.5}$ for the whole city, a considerable part of the cutting contribution was from point sources. Compared to area sources, point sources had much less effect on air pollutants at the ground level. Besides, the emission reduction was not even. All of these could result in less effect on reducing pollutants in simulation. For primary particles, emission abatement independently led to a 39.6% decrease in $PM_{10p}$ and a 26.2% decrease in $PM_{2.5p}$. For secondary particles, emissions of $SO_2$, $NO_x$ and $NH_3$ contribute to the production of sulfate, nitrate and ammonium salt, respectively. And no control in the emission of $NH_3$ could weaken the reduction effect on $PM_{10s}$ and $PM_{2.5s}$.

The details about setting of the simulation schemes were in Section 2. The emission reduction simulation schemes were based on the local government emission controls during the 2nd YOG. If we increase the intensity of emission reduction, of course the effect of emission abatement will be more obvious, but it is not reasonable for discovering the influence factor during the 2nd YOG. 2013 is a normal meteorological year, and it is reasonable to be used in the simulation. Thus, we think it unnecessary to use different scales for the changes from meteorology and emission reduction.

[Figure]

Fig. 12. The simulated effect of meteorology and reduction on pollutant concentrations in Nanjing during 1-31 Aug. , 2014, Met. (Exp.2-Exp.3) represents the effect of meteorology, while Red. (Exp.1-Exp.2) represents the simulated effect of reduction.

**Table 6**

Comparison between the simulated effect of meteorology and emission reduction at CCM and XL station

| Species | Met. (CCM) | Red. (CCM) | Met. (XL) | Red. (XL) | Met. (NJ) | Red. (NJ) |
|---|---|---|---|---|---|---|
| $SO_2$ | 17.4% | -24.3% | 14.1% | -19.2% | 17.5% | -24.6% |
| $NO_2$ | 15.1% | -11.7% | 12.4% | -10.2% | 16.9% | -12.1% |
| $PM_{10}$ | 15.6% | -13.9% | 22.4% | -11.9% | 18.5% | -15.1% |
| $PM_{2.5}$ | 14.9% | -7.5% | 24.5% | -6.3% | 18.8% | -8.1% |
| CO | 6.4% | -7.0% | 2.3% | -5.5% | 7.8% | -7.2% |
| $O_3$ | 0.9% | 1.3% | 1.6% | 0.9% | 0.7% | 1.5% |
| $PM_{10p}$ | 13.2% | -38.3% | 5.9% | -33.2% | 12.6% | -39.6% |
| $PM_{10s}$ | 16.7% | -2.4% | 29.4% | -2.9% | 21.5% | -2.9% |
| $PM_{2.5p}$ | 8.4% | -25.8% | 4.9% | -20.1% | 9.5% | -26.2% |
| $PM_{2.5s}$ | 16.7% | -2.4% | 29.4% | -2.9% | 21.5% | -2.9% |

Met.: the change percentage of species in Exp.2 based on Exp3, represents the effect of meteorology.
Red.: the change percentage of species in Exp.1 based on Exp 2, represents the effect of Nanjing local emission reduction.

**Comment 7:** It's OK to keep the tables if the authors think they are necessary.

**Response:** Thank you for your comment.

**Comment 8:** The authors clarify the questions.

**Response:** Thank you for your comment.

**Reviewer 3**

**Comment 1:** The authors have made substantial improvement of this version of the manuscript, and most of the reviewer's comments were addressed. The quality of the figures and the language need technical corrections. For example, the marks in Fig.1 is not clear.

**Response:** Thank you for your comment. We have checked and revised some figures including Fig. 1. Besides, the co-authors have helped to modify and improve the English in the manuscript carefully.

**Change list:**

**1. Line 5-14:** Adjust the affiliation of the first author.

**2. Line 29, 32, 34, 36:** Revise the Abstract.

**3. Line 58, 60, 61, 64, 89, 99, 100, 102, 106, 107:** Revise Section 1 Introduction.

**4. Line 113, 115, 126-127, 134, 153, 155-158:** Revise Section 2 Methodology.

**5. Line 150-163:** Revise Fig. 1, and make it more clear.

**6. Line 165-166:** Revise the caption of Table 1.

**7. Line 174-176, 183:** Rephrase Section 2.3 Emissions and simulation scenarios.

**8. Line 192:** Adjust the arrangement of pictures in Fig. 2.

**9. 196, 200-205:** Rephrase Section 2.3 Emissions and simulation scenarios.

**10. Line 208, 223:** Rephrase Section 3.1 Observed air quality during the YOG.

**11. Line 228-229, 233-234:** Revise the caption of Fig. 3 and Fig. 4.

**12. Line 248, 255, 266-267:** Rephrase Section 3.1.

**13. Line 289-290:** Revise the caption of Table 5.

**14. Line 296, 300-339:** Revise Section 3.2 Simulated impact of meteorological conditions.

**15. Line 341-342:** Revise Fig. 7 and the caption of it.

**16. Line 347-351:** Add a figure (Fig. 8) to show the influence of meteorology on hourly mean concentrations of primary and secondary particulate matters in Aug. 2014 compared with Aug. 2013.

**17. Line 353-359:** Add more meteorological factors (relative humidity at 2m , cloud fraction, and net downward short wave flux at ground surface) in Fig. 9.

**18. Line 370, 373-378:** Revise Section 3.3 Simulated impact of emission reduction.

**19. Line 379-380:** Revise Fig. 10 and the caption of it.

**20. Line 385-390:** Add a figure (Fig. 11) to show the influence of emission reduction on hourly mean concentrations of primary and secondary particulate matters in Aug. 2014.

**21. Line 393-400:** Rephrase Section 3.4 Comparison of simulated meteorological factors and emission reduction.

**22. Line 402-405:** Change Fig. 12 and add more species ($PM_{10p}$, $PM_{10s}$, $PM_{2.5p}$, and $PM_{2.5s}$) in it.

**23. Line 407-431:** Rephrase Section 3.4.

**24. Line 433-438:** Chang Table 6 and add more species ($PM_{10p}$, $PM_{10s}$, $PM_{2.5p}$, and $PM_{2.5s}$) in it.

**25. Line 443-452:** Rephrase Section 3.4.

**26. Line 457-464, 467-473, 475-477:** Rephrase Section 4 Summary and conclusions.

**27. Add some references as listed below:**

Katragkou E., Zanis P., Kioutsioukis I., Tegoulias I., Melas D., Krüger, B. C., and Coppola E.: Future climate change impacts on summer surface ozone from regional climate-air quality simulations over Europe, Journal of Geophysical Research Atmospheres, 116, D22307, doi:10.1029/2011JD015899, 2011.

Pu X., Wang T. J., Huang X., Melas D., Zanis P., Papanastasiou D. K., and Poupkou A.: Enhanced surface ozone during the heat wave of 2013 in Yangtze River Delta region, China. Science of the Total Environment, 603-604, 807-816, doi:10.1016/j.scitotenv.2017.03.056, 2017.

**28.** Revise the figure and table format to be uniform.

**29.** Correct the grammar and spelling mistakes throughout the manuscript.

[revised manuscript text omitted]

---

## Author Response (AR3)

Dear Editors and Reviewers,

Thank you very much for your letter and for the reviewers' comments concerning our manuscript entitled "Impacts of emission reduction and meteorological conditions on air quality improvement during the 2014 Youth Olympic Games in Nanjing, China" (doi:10.5194/acp-2017-114). Your comments are all valuable and very helpful for revising and improving our paper, as well as the important guiding significance to our researches. We have investigated the comments carefully and made corrections which we hope to meet with approval. Based on the instructions, we have uploaded the file of the revised manuscript.

Appended to this letter is our point-by-point response to the reviewers' comments, the change list and the marked-up manuscript.

We would like to thank you for allowing us to resubmit a revised copy of the manuscript. We hope that the revised manuscript can be accepted for publication in ACP.

Sincerely,

Qian Huang

**Responses to the reviewers' comments:**

**Reviewer 1**

**Comment 1:** The authors have largely improved the manuscript by addressing my comments and revising the manuscript accordingly, most of the questions have been clarified and additional model simulations were conduced and interpreted, with some new interesting results presented. I recommend acceptance of this manuscript after minor revisions on English and tables/figures.

**Response:** Thank you very much for your comment. We have carefully checked and modified the tables and figures, like the legend of Fig. 5 and Fig. 6, to make them more standardized. And we have checked the grammar and revise some sentences. Prof. Wang and Doctor Zhuang also helped to check and improve the English.

**Change list:**

**1. Line 23-24:** Revise the abstract.

**2. Line 52-53, 61, 63, 71, 76, 87, 93-94, 103:** Revise Section 1 Introduction.

**3. Line 119, 121, 127, 131, 148-150, 158:** Revise Section 2 Methodology.

**4. Line 167:** Revise Table 1.

**5. Line 171, 173, 188-191:** Revise Section 2.3 Emissions and simulation scenarios.

**6. Line 194:** Revise the caption of Fig. 2.

**7. Line 196-197, 204:** Revise Section 2.3.

**8. Line 208, 210-212, 215-218, 223-225:** Revise Section 3 Results and discussions.

**9. Line 239-240:** Revise Table 2 and its caption.

**10. Line 244-245:** Revise Table 3 and its caption.

**11. Line 247, 260-261, 263:** Revise Section 3 Results and discussions

**12. Line 271:** Revise the legend of Fig. 5, and make it more clear.

**13. Line 276:** Revise the legend of Fig. 6, and make it more clear.

**14. Line 283-285:** Revise Table 4 and its caption.

**15. Line 289-291:** Revise Table 5 and its caption.

**16. Line 305, 309, 320-324, 366, 395-397:** Revise Section 3.

**17. Line 403-405:** Revise the caption of Fig. 12.

**18. Line 407-411, 418:** Revise Section 3.4.

**19. Line 436-437:** Revise the caption of Table 6.

**20. Line 440-441, 443-444, 449, 451-452:** Revise Section 3.4.

**21. Line 466:** Revise Section 4.

**22. Line 480-487:** Add Section 5 Data availability and declare about competing interests.

**23. Line 491:** Revise the acknowledgements.

[revised manuscript text omitted]